# Heterogeneously integrated lithium tantalate-on-silicon nitride modulators for high-speed communications

Jiachen Cai [1,2,5], Alexander Kotz [3,5], Hugo Larocque [2,5], Chengli Wang[1,2], Xinru Ji [2], Junyin Zhang [2], Daniel Drayss[3], Jiale Sun[2], Shuhang Zheng[2], Xin Ou [1] ✉, Christian Koos [3] ✉ & Tobias J. Kippenberg [2,4] ✉

Ultrabroadband integrated modulators involving materials beyond those available in silicon manufacturing increasingly rely on the Pockels effect. Among electro-optic materials, lithium tantalate offers comparable Pockels coefficients to lithium niobate but with significantly improved photostability, lower birefringence, higher optical damage threshold, and enhanced DC bias stability. Here we demonstrate wafer-scale heterogeneous integration of lithium tantalate films on low-loss silicon nitride photonic integrated circuits, achieving low optical losses ( ~ 14.2 dB/m) while combining the mature processing of silicon nitride waveguides with the ultrafast electro-optic response of thin-film lithium tantalate. The resulting devices achieve a 6 V half-wave voltage, and support modulation bandwidths of up to 100 GHz. We use single intensity modulators and in-phase/quadrature (IQ) modulators to transmit PAM4 and 16-QAM signals reaching up to 333 and 581 Gbit/s net data rates, respectively. Our results establish lithium tantalate-on-silicon nitride as a viable platform for RF photonics, interconnects, and analog signal processing.

Photonic integrated circuits (PICs) provide a unique approach to scaling standardized optoelectronic technologies[1,2]. Among PIC platforms, silicon nitride-based devices offer a range of features traditionally leveraged in optical fibers such as low propagation losses, large power handling[3], complementary metal-oxide-semiconductor (CMOS) compatibility, and a wide bandgap enabling transparency across a wide range of optical wavelengths[4–6]. Thicker nitride waveguides, e.g., manufactured with a photonic Damascene process, additionally provide strong optical confinement and easily-achievable anomalous group velocity dispersion[7–10]. These features become crucial while harnessing this material's optical nonlinearities[11] and have thus lead to demonstrations pertaining to Kerr frequency combs[10–13], optical frequency conversion[14,15], traveling-wave optical parametric amplifiers[16–18], and quantum information science[19–22]. Further deployment of this platform in applications such as optical communications and microwave photonics[23,24] requires electro-optic modulation, which silicon nitride cannot directly provide due to its amorphous nature.

Ferroelectric-based PICs readily provide such features[25–27], and can thus introduce electro-optic modulation capabilities into silicon nitride PICs through the heterogeneous integration of materials such as lithium niobate[28,29]. Recent advances that have demonstrated low-loss lithium tantalate PICs[30], as well as its advantages such as lower birefringence, lower bias drift, and higher power handling than in LiNbO₃, motivate applying similar integration methods to alternative Pockels materials[31,32]. Improved economies of scale for lithium tantalate substrates due to their role in 5G/6G RF filters[33] further encourage

[1]State Key Laboratory of Materials for Integrated Circuits, Shanghai Institute of Microsystem and Information Technology, Chinese Academy of Sciences, Shanghai, China. [2]Institute of Physics, Swiss Federal Institute of Technology Lausanne (EPFL), Lausanne, Switzerland. [3]Institute of Photonics and Quantum Electronics (IPQ), Karlsruhe Institute of Technology (KIT), Karlsruhe, Germany. [4]Institute of Electrical and Micro engineering, Swiss Federal Institute of Technology, Lausanne (EPFL), Lausanne, Switzerland. [5]These authors contributed equally: Jiachen Cai, Alexander Kotz, Hugo Larocque. ✉e-mail: ouxin@mail.sim.ac.cn; christian.koos@kit.edu; tobias.kippenberg@epfl.ch

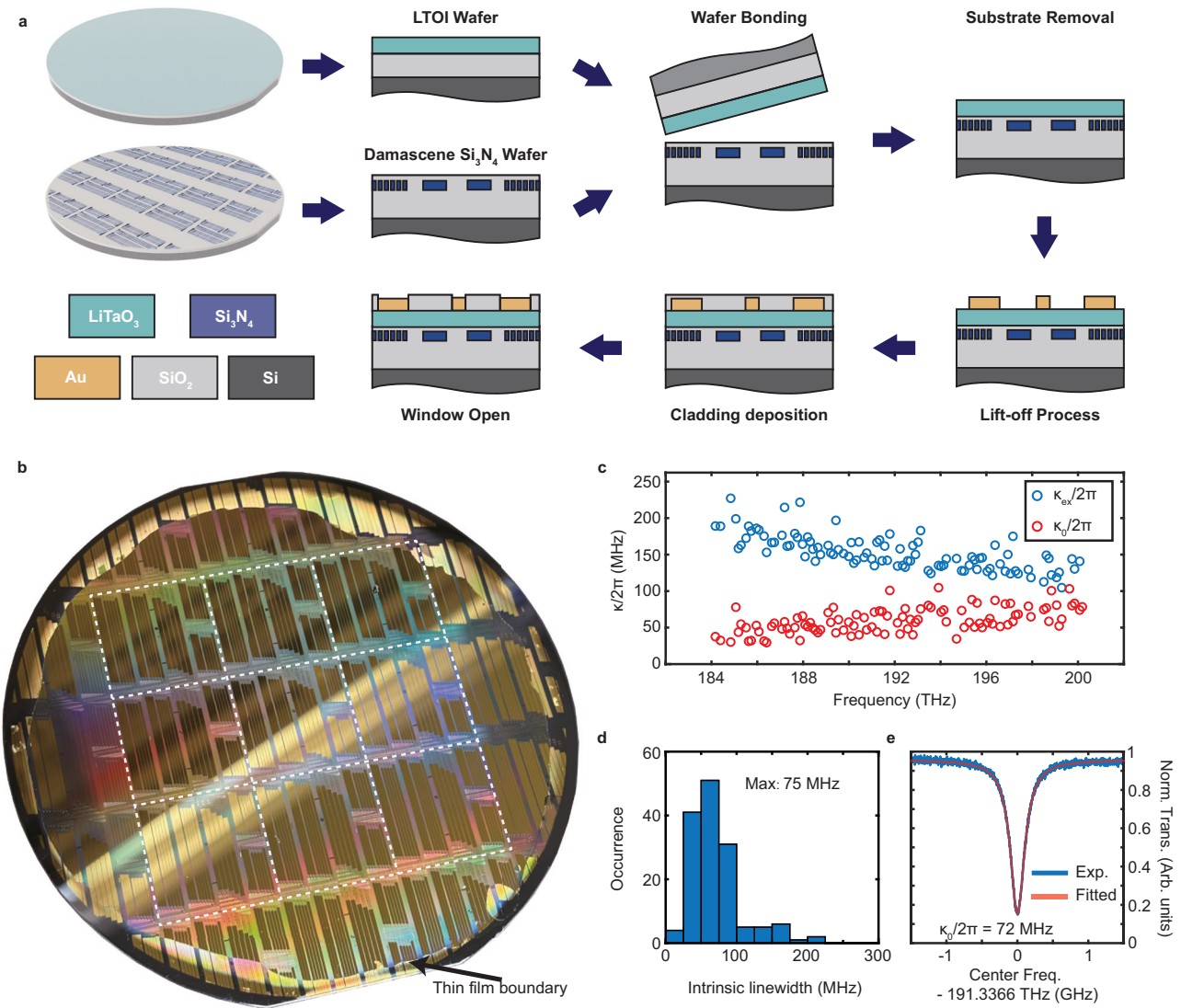

**Fig. 1 | Wafer-scale manufacturing of $Si_3N_4$-$LiTaO_3$ photonic integrated circuits.**
**a** Schematic diagram of the main steps in the fabrication of heterogeneously integrated $Si_3N_4$-$LiTaO_3$ photonic circuits. **b** Photograph showing the 100 mm $Si_3N_4$-$LiTaO_3$ wafer with high fabrication yield (wafer ID: D142_LT). **c** Fitted value of external linewidth, $\kappa_{ex}$, and intrinsic linewidth, $\kappa_0$, for a ring resonator with a 2 μm

waveguide width and 112 GHz free spectral range. **d** Corresponding fitted $\kappa_0$ histogram. The maximum occurrence at 75 MHz suggests that most of the resonances' intrinsic linewidths lie between 62.5 and 87.5 MHz. **e** Representative normalized resonance with an intrinsic linewidth of 72 MHz.

the use of this material in integrated modulators. At the current stage, heterogeneous integration of this material offers an immediately accessible route towards leveraging its features in high-volume manufacturing platforms such as silicon nitride.

Here, we develop a wafer-scale bonding process for lithium tantalate on silicon nitride PICs. With this platform, we demonstrate modulation defined by a $V_\pi L = 4.08 V \cdot cm$ and a 3-dB bandwidth close to 100 GHz in a push–pull Mach–Zehnder modulator (MZM). These results enable net data transmission rates of 333 Gbit/s when operating the modulators in pulse amplitude modulation (PAM) schemes. We further extend this data transmission demonstration to $Si_3N_4$-$LiTaO_3$ IQ modulators, thereby achieving net data transmission rates of 581 Gbit/s using quadrature amplitude modulation (QAM) signals. With performance metrics that can already compete with those of monolithic lithium-tantalate-on-insulator (LTOI) PICs[31,32], these results show that $Si_3N_4$-$LiTaO_3$ devices can implement systems combining mature $Si_3N_4$ PICs with next-generation ferroelectric thin films, thereby enabling both low-loss strong optical confinement with GHz-rate modulation.

## Results

### Hybrid $Si_3N_4$-$LiTaO_3$ PIC fabrication

Figure 1a outlines the process flow for the hybrid PICs, which starts with the fabrication of $Si_3N_4$ waveguide structures using the photonic Damascene process[9]. As detailed in Supplementary Note 1, the process flow begins with preform formation on a 100 mm-diameter silicon wafer with 4 μm thick wet thermal oxide by means of deep ultraviolet stepper lithography, dry etching, and reflow under high temperatures. Low-pressure chemical vapor deposition (LPCVD) then deposits a layer of $Si_3N_4$, which subsequently undergoes chemical-mechanical polishing, oxide interlayer deposition, and annealing. The $Si_3N_4$ photonic Damascene process is free of crack formation in the highly tensile LPCVD $Si_3N_4$ film and provides high fabrication yields and low optical propagation losses of $\mathcal{O}(dB/m)$[8]. The patterned sample supports strip waveguides with a 1 μm width and a 0.5 μm height. In addition, inverse nanotapers allow for efficient edge coupling between the chip's waveguides and lensed fibers.

Our heterogeneous integration approach additionally involves 100 mm LTOI wafers fabricated by a hydrogen-based ion-slicing

technique[30,33]. The resulting wafer stack consists of a 300 nm x-cut lithium tantalate thin film, a 2 μm buried oxide layer, and a 525 μm thick silicon substrate. Following surface preparation relying on cleaning and plasma activation, the $Si_3N_4$ and LTOI wafers undergo hydrophilic pre-bonding at room temperature, where van der Waals interactions establish a preliminary bond between the two wafers. A subsequent 300°C thermal annealing process then enhances the bond strength by accelerating the polymerization of silanol (Si-OH) groups to covalent bonds (Si-O-Si) at the interface.

After bonding, the process continues by thinning the $LiTaO_3$-based donor substrate. Backside wafer grinding followed by a tetramethylammonium hydroxide wet etch entirely removes the substrate's silicon. A buffered hydrofluoric acid solution then eliminates the thermal oxide layer, thereby exposing the $LiTaO_3$ layer and forming a well-confined hybrid optical waveguide. This sequence entirely lacks plasma-based processing traditionally used in $LiTaO_3$ nanofabrication, which for instance includes oxygen-based diamond-like carbon hardmask etching[34] and argon dry etching[35]. With such measures, we avoid introducing potential plasma-induced charges at the $Si_3N_4$ PIC wafer's $Si/SiO_2$ interface, which can lead to parasitic surface conduction RF losses[36].

To add high-speed coplanar waveguide (CPW) and heater electrodes to the sample, lithography based on a maskless aligner (Heidelberg Instruments MLA150) first defines the electrode layout. Thermal evaporation of 10 nm of titanium and 800 nm of gold, followed by lift-off, completes the electrode's fabrication. Argon-based ion beam etching subsequently patterns the $LiTaO_3$ thin film to form adiabatic taper transitions between $Si_3N_4$ waveguides and the hybrid $Si_3N_4$-$LiTaO_3$ waveguides. The process also fully removes the $LiTaO_3$ near the chip facets to improve edge coupling efficiency to the hybrid $Si_3N_4$-$LiTaO_3$ PIC. After cladding deposition, an additional exposure step followed by hydrofluoric wet etching selectively expose portions of the sample for landing probes on the manufactured electrodes. As discussed in Supplementary Note 2, these cladding openings also match the group velocities of the CPW RF field and optical waveguide mode.

Figure 1b shows a photograph of a finalized 100-mm hybrid $Si_3N_4$-$LiTaO_3$ wafer, where a distinct boundary line delineates the contour of the bonded $LiTaO_3$ film. The central nine stepper fields are fully-covered by the film, thereby highlighting the excellent fabrication consistency and large-scale yield of the wafer bonding technique. Supplementary Note 3 provides further details on wafer-scale yields, which include film thickness homogeneity and device performance analyses.

To characterize the optical properties of the wafer's hybrid photonic structures, frequency-comb-assisted spectroscopy[37] relying on three external-cavity diode lasers probed optical waveguide loss in fabricated hybrid $Si_3N_4$-$LiTaO_3$ microring resonators. Figure 1c, d show the derived external, $\kappa_{ex}$, and intrinsic, $\kappa_0$, loss rates of the ring's TE polarization resonances near the C telecommunication band. Here, $\kappa_0$ increases with optical frequency, which is consistent with the behavior of similar structures implemented in $LiNbO_3$-on-$Si_3N_4$ heterogeneous PICs[28,29]. In such structures, higher optical frequencies result in a smaller mode field defined by a stronger overlap with the higher-index ferroelectric slab, thus inducing additional bending losses in the rings. Over the frequency range considered in Fig. 1d, finite element method simulations suggest the lithium tantalate film holds roughly 48% of the mode's energy. Figure 1e shows a typical normalized resonance in the C-band, which indicates a fitted intrinsic linewidth of 72 MHz and a corresponding propagation optical loss of $\alpha \approx 14.2$ dB/m. This figure is larger than the 3–4 dB/m and 5.6 dB/m loss values achieved in monolithic $Si_3N_4$[7,8] and $LiTaO_3$[30] circuits. This comparison provides further evidence that the hybrid waveguide's losses do not come from intrinsic material losses but rather radiative losses in the unpatterned and higher-index $LiTaO_3$ film.

## Electro-optic performance

To demonstrate electro-optic capabilities in our hybrid $Si_3N_4$-$LiTaO_3$ platform, we rely on electro-optic modulators consisting of 6.8 mm-long MZMs operating in a push-pull configuration with high-speed CPW electrodes. Figure 2a shows a micrograph of such a modulator. Supplementary Note 4 provides simulated data regarding the transmission of some of its underlying components. The CPW design features a signal-to-ground spacing that accounts for trade-offs between metal-induced optical losses and modulation. From the cross-sectional SEM images in Fig. 2b and the inset diagram in Fig. 2c, our CPW design balances out these two factors with a 23 μm signal electrode width and a 6 μm gap separating the ground and signal electrodes. Simulations presented in Supplementary Note 2 suggest this configuration yields metal-induced propagation losses of 0.1 dB/cm and a voltage length product of $V_\pi L \sim 4.5 \, V \cdot cm$. As further discussed in Supplementary Note 2, the chosen signal width also affects the modulator's underlying bandwidth by altering RF propagation losses and velocity mismatching between the modulator's co-propagating RF and optical fields.

We first characterize the modulator's electro-optic performance under quasi-DC operation. Coupling 1550 nm continuous-wave light into the $Si_3N_4$-$LiTaO_3$ MZM exhibited a fiber-to-chip coupling loss of approximately 5 dB. Figure 2c shows the measured output power while applying a 100 Hz triangle voltage wave via an RF probe. These results indicate a $V_\pi = 6V$ half-wave voltage for a 6.8 mm long push-pull configuration, which corresponds to a $V_\pi L = 4.08 V \cdot cm$ voltage-length product. Additional data provided in Supplementary Note 5 suggest this response is consistent down to modulation frequencies of at least 1 Hz and over optical wavelengths ranging from 1500 to 1630 nm.

As highlighted in Fig. 2d, probing the modulator's response with a vector network analyzer (VNA) provided the modulator's bandwidth. First, a two-port calibration using short-open-load-through standards on a commercial calibration substrate (AC2-2, MPI Corporation) is performed, which sets the reference plane at the tips of the two high-speed electrical probes. The electro-optic-electric measurement then relies on a modification in the setup where one of the VNA's outputs connects to the high-speed photodiode detecting the optical output, rather than to the electrical probe. De-embedding independently characterized frequency responses from the photodiode and one electrical probe finally yields the intrinsic electro-optic (EO) response of the modulator. The resulting $S_{21}$ EO response exhibits excellent flatness within 3 dB across frequencies ranging from 25 MHz to 110 GHz while keeping $S_{11}$ microwave reflections below −15 dB. These metrics reach levels comparable to state-of-the-art figures achieved in other electro-optic PIC platforms[31,38,39], which underscores the significant potential of $Si_3N_4$-$LiTaO_3$ for applications requiring high-speed electro-optic modulation.

Monitoring the modulator's optical output power while set to its quadrature point quantified its stability under a DC bias. As outlined in Supplementary Note 6, a dedicated photonic package mitigated the influence of drifting chip-to-fiber coupling efficiency in this measurement. As shown in Fig. 2e, the $Si_3N_4$-$LiTaO_3$ MZM exhibits low power shifts of less than 0.5 dB over a 1-h period, thus outperforming results reported in prior work on monolithic lithium niobate and lithium tantalate PIC platforms[38,40,41]. Minimal degradation of the unetched $LiTaO_3$ film, where defect-driven photorefractive effects typically give rise to drift[42], likely contributes to the modulator's enhanced DC bias stability.

## Data transmission experiments

To demonstrate the viability and performance of our $Si_3N_4$-$LiTaO_3$ platform, we use our devices in high-speed optical data communication experiments, covering both intensity-modulation and direct-detection (IMDD) and coherent modulation schemes. Figure 3a shows the setup for the IMDD experiment. Herein, a tunable external-cavity laser (ECL) operating in the C-band with a power of 17.8 dBm provides the optical

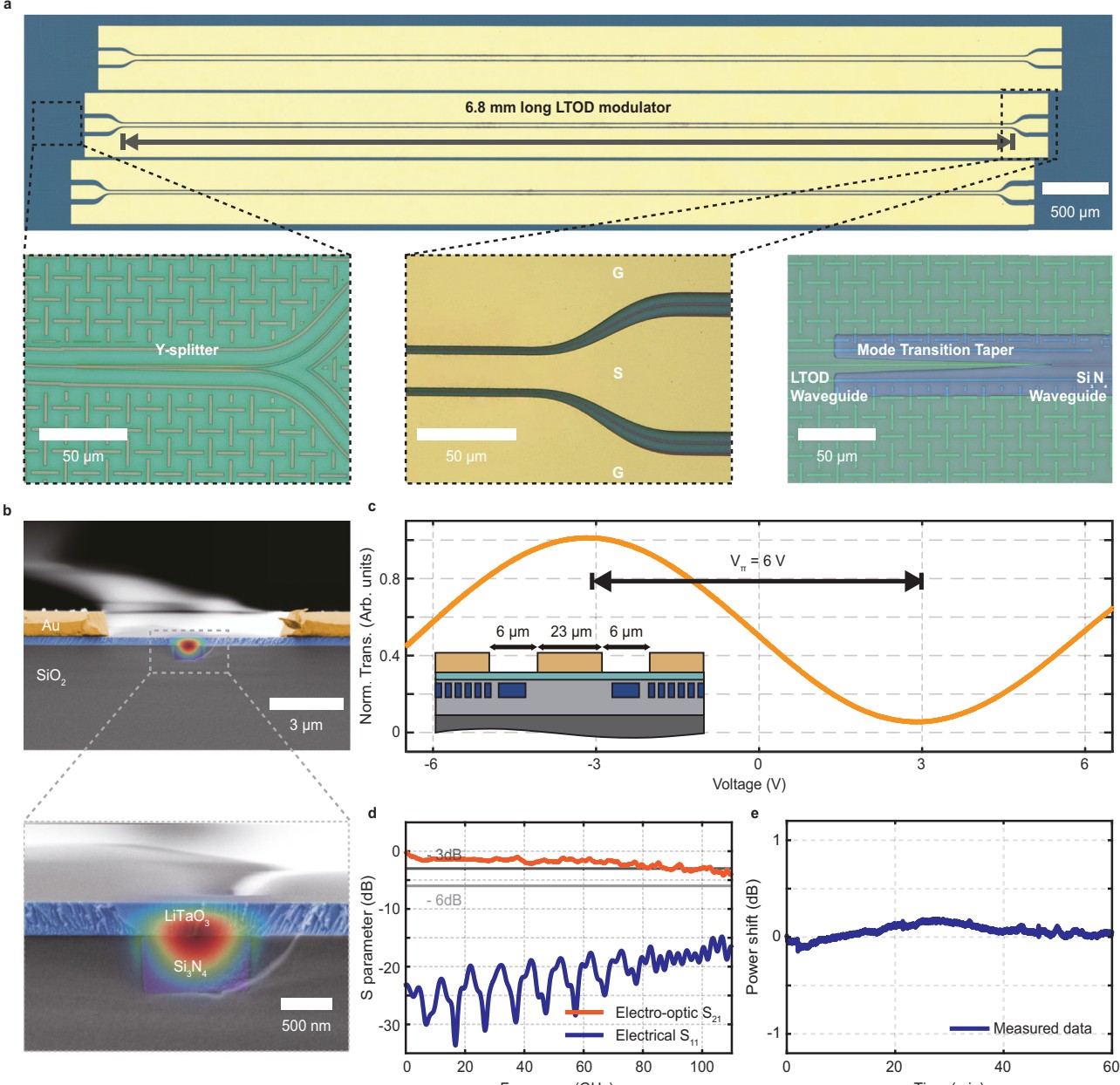

**Fig. 2 | Hybrid Si$_3$N$_4$-LiTaO$_3$ electro-optic modulators. a** Optical micrograph of a fabricated 6.8 mm-long hybrid modulator. Insets: optical micrographs of the modulator's underlying components, which include Y-splitters, a coplanar waveguide electrode, and tapered transitions between Si$_3$N$_4$ waveguides and Si$_3$N$_4$-LiTaO$_3$ waveguides (LTOD) with a simulated 0.28 dB insertion loss. **b** False-colored scanning electron microscopy (SEM) image of the cross-section of a manufactured Si$_3$N$_4$-LiTaO$_3$ modulator. **c** Normalized transmission of a 6.8 mm-long push-pull MZM versus applied voltage. The inset diagram shows the high-speed electrode geometry with a 23 µm signal width and a 6 µm electrode gap. **d** Measured electro-optic response (electro-optic S$_{21}$) and microwave return loss (electrical reflection S$_{11}$), revealing a high 3-dB bandwidth near 100 GHz. **e** Output optical intensity of the modulator while biased at its working voltage, i.e., its quadrature point, over 1 h.

carrier. A fiber-based polarization controller (FPC) then adjusts the output polarization before coupling to the quasi-TE mode, having a dominant electric field parallel to the substrate plane, of the Si$_3$N$_4$-LiTaO$_3$ MZM via a pair of lensed fibers. We drive the MZM with electrical PAM4 signals that are synthesized by offline digital signal processing (Tx-DSP) based on pseudo-random bit sequences and root-raised cosine pulse-shaping filters[31], and that are converted to the analog domain using a high-speed arbitrary waveform generator (AWG, M8199B, Keysight Technologies Inc.). A 20 cm-long RF cable followed by a broadband RF amplifier (SHF T850 B, SHF Communication Technologies AG) sends the drive signals to the coplanar transmission line of the MZM via a first impedance-matched probe in a ground-signal-ground

configuration. A second probe terminates the transmission line with a 50 Ω coaxial termination. Tx-DSP compensates for frequency-dependent RF loss up to the input of the feeding probe by implementing a linear minimum-mean-square-error predistortion. An erbium-doped fiber amplifier (EDFA) brings the rather weak output signal of −10 dBm from the MZM to a power level near 10 dBm compatible with the receiver. This receiver comprises a high-speed photodiode (PD, Finisar Corp.), which is directly connected to a high-speed real-time oscilloscope (RTO, UXR 1004A, Keysight Technologies Inc.) with a sampling rate of 256 GSa/s and an analog bandwidth of 105 GHz. The EDFA is followed by an optical bandpass filter (BPF) to suppress out-of-band amplified spontaneous-emission (ASE) noise and by a variable

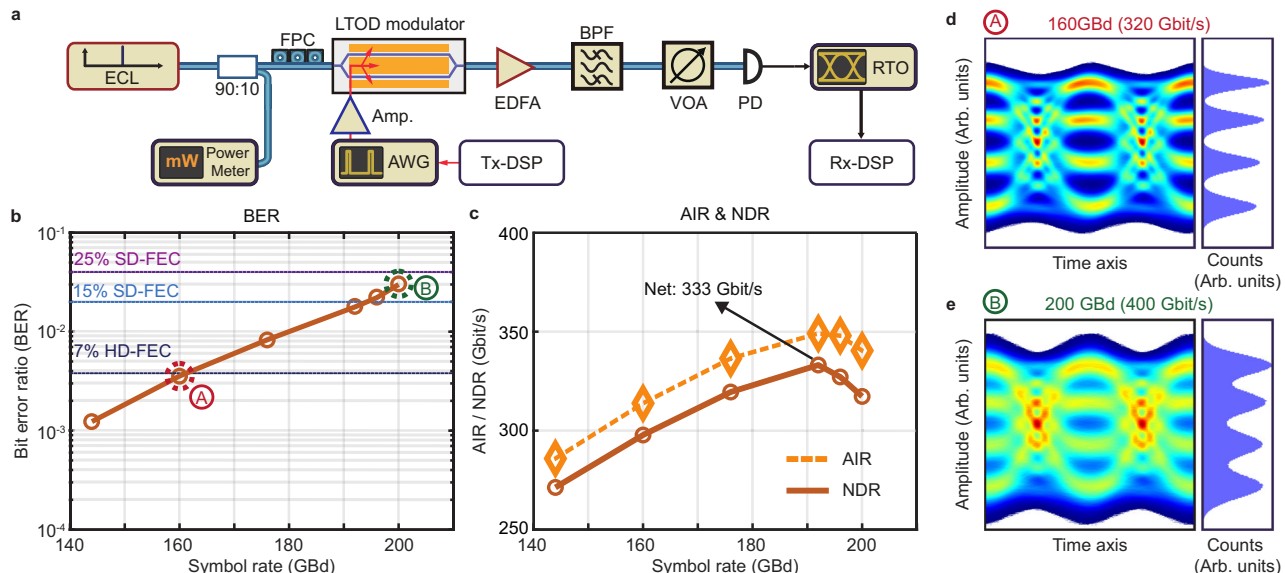

**Fig. 3 | Transmission experiments using intensity-modulation and direct detection (IMDD). a** Experimental setup. ECL External cavity laser, FPC Fiber polarization controller, AWG Arbitrary-waveform generator, Tx-DSP Transmitter-digital signal processing (offline), Amp RF amplifier, LTOD Si₃N₄-LiTaO₃ modulator, EDFA Erbium-doped fiber amplifier, BPF Bandpass filter, VOA Variable optical attenuator, PD Photodiode, RTO High-speed real-time oscilloscope (256 GSa/s, 105 GHz), Rx-DSP Receiver-digital signal processing (offline). **b** Measured pre-forward error correction bit error ratio (BER) versus symbol rate for PAM4 transmission, with horizontal dashed lines indicating the thresholds for soft-decision forward error correction with 15% and 25% coding overhead (15% SD-FEC, 25% SD-FEC) and for hard-decision FEC with 7% coding overhead (7% HD-FEC). The eye diagrams of two selected data points are shown in (**d**, **e**). **c** Achievable information rates (AIR, orange dashed line) and corresponding net data rates (NDR, solid brown line) of the IMDD measurements, showing that the highest achieved NDR is 333 Gbit/s using a PAM4 signals at a symbol rate of 192 GBd. Eye diagrams (left) at selected symbol rate of 160 GBd PAM4 (**d**) and 200 GBd PAM4 (**e**), labeled in (**b**), along with the associated histograms (right) of the reconstructed signal amplitudes in the center of the corresponding symbol slot.

optical attenuator (VOA, LTB-1, EXFO Inc.) that adjusts the optical power to 10 dBm, the maximum input power accepted by the photodiode. At the receiver, we use offline digital signal processing (Rx-DSP) to extract and demodulate the PAM4 data. As further elaborated in Supplementary Note 7, the Rx-DSP suite comprises standard algorithms such as timing recovery, linear Sato equalization, and an additional decision-directed least-mean-square equalizer.

In our experiment, we used this setup to operate the device at different PAM4 symbol rates between 144 GBd and 200 GBd. Figure 3b provides the resulting pre-forward error correction (pre-FEC) bit error ratios (BER) along with the BER thresholds for different forward error correction (FEC) schemes indicated by dashed horizontal lines. For symbol rates of 160 GBd (line rate 320 Gbit/s) and lower, the BER is below the threshold for hard-decision FEC with 7% coding overhead (7% HD-FEC)[43], and it stays below the threshold for soft-decision FEC (SD-FEC) with 15% coding overhead [44, Table 7.5] up to symbol rates of 192 GBd (line rate 384 Gbit/s). For 200 GBd (line rate 400 Gbit/s), we find a BER value that is still compatible with soft-decision forward-error correction (SD-FEC) with 25% coding overhead (25% SD-FEC) [44, Table 7.5]. Figure 3d, e depict the reconstructed eye diagrams of the 160 GBd and the 200 GBd signals, respectively, along with the histograms taken at the center of the symbol slot. The data points corresponding to these eye diagrams are marked in Fig. 3b. While the signal quality clearly leaves room for improvement, the demonstrated symbol rates and line rates can already compete with those achieved by standalone LTOI MZM[31].

To quantify the information transfer efficiency of our transmission system, we derive the generalized mutual information (GMI) of the transmitted signals from the log-likelihood ratios of the received symbols based on an additive white Gaussian noise channel model[45]. Figure 3c shows the resulting achievable information rates (AIR), which correspond to the product of the symbol rate and the GMI of each symbol and provide an upper bound for the transmission capacity of

the system. To estimate practically achievable net data rates (NDR), we additionally have to include penalties introduced by typical FEC codes. By comparing the normalized GMI (NGMI) estimated in our experiments (see Supplementary Note 8) to the NGMI thresholds provided in ref. 46, we select suitable FEC codes and evaluate the NDR by multiplying the line rates and the associated FEC code rates. The highest NDR of 333 Gbit/s is obtained for a symbol rate of 192 GBd (line rate 384 Gbit/s) in combination with 15% SD-FEC, while the AIR at this symbol rate amounts to 349 Gbit/s.

Besides MZM and IMDD signaling, our Si₃N₄-LiTaO₃ platform also supports more advanced systems, such as IQ modulators (IQM) for coherent communications. Figure 4a shows a micrograph of such an IQM. The device consists of an input waveguide followed by a Y-splitter leading to two 13.5 mm-long MZMs, such as the one shown in Fig. 2a. The individual MZMs are DC-biased at their minimum transmission point using two 110 GHz bias-Tees (BT110R-C, SHF Communication Technologies AG). A heater-based phase shifter, which can produce a drift-free thermo-optic phase shift[47], then sets the relative phase between the signals generated in the in-phase (I) and the quadrature (Q) arms of the IQM to 90°. Thereafter, a power combiner, like the one in Fig. 2a, synthesizes the output signal with two modulated quadratures. To demonstrate the viability of the device, we use the setup shown in Fig. 4b to transmit quadrature phase-shift keying (QPSK) and 16-state quadrature-amplitude modulation (16QAM) signals at symbol rates between 144 and 204 GBd. The setup shares several similarities with the one used for IMDD transmission shown Fig. 3a. Here, the two AWG channels, however, individually drive the I and Q arms of the IQM. Supplementary Note 8 gives further details on the drive voltages used in this demonstration. After modulation, an EDFA amplifies the optical signal from −13 to 16 dBm, and subsequent bandpass filting reduces out-of-band spontaneous emission noise. At the receiver, the modulated signal enters a 90° optical hybrid module, where it interferes with a

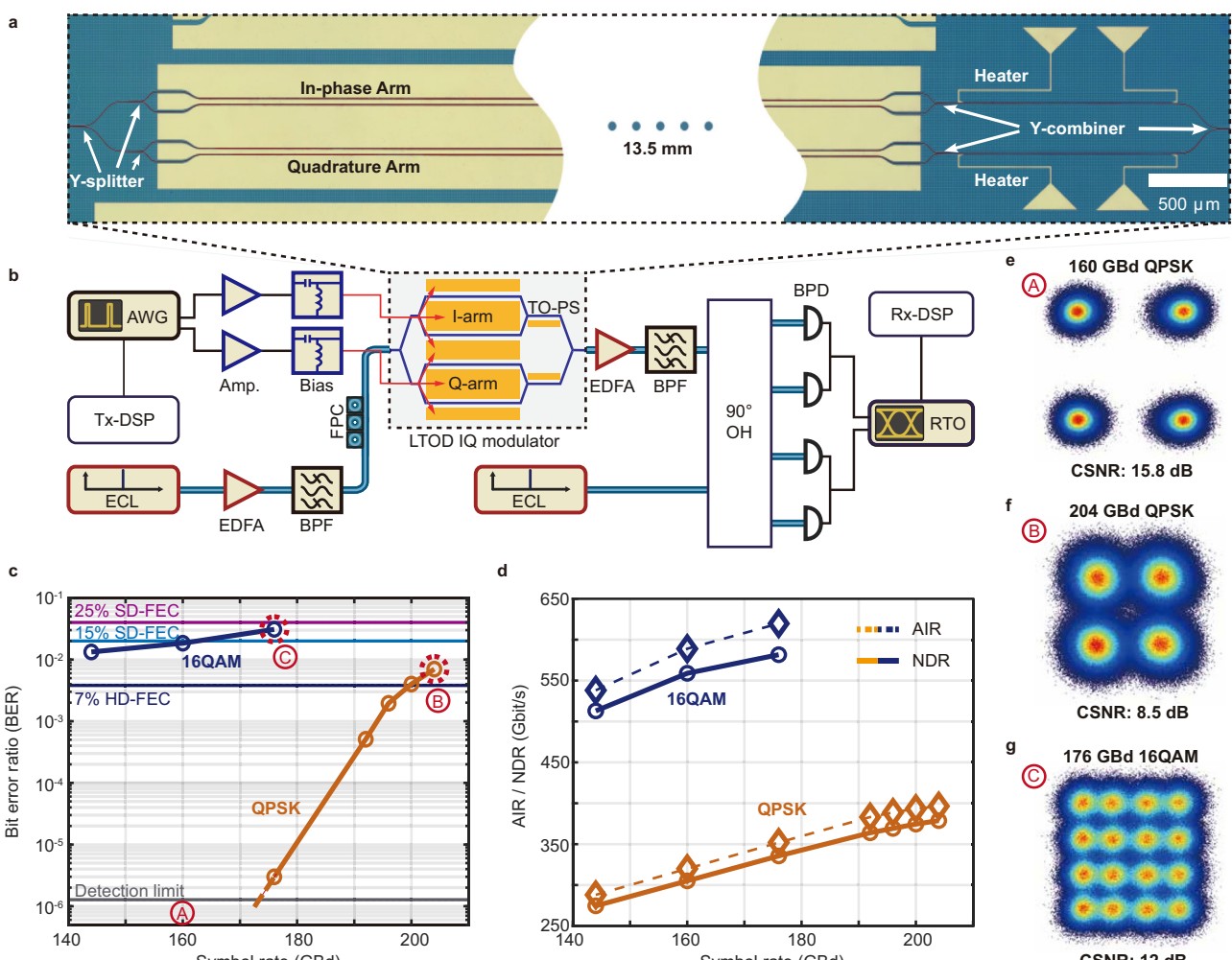

**Fig. 4 | Hybrid Si₃N₄-LiTaO₃ IQ modulator for coherent data transmission.**
**a** Optical micrograph of a 13.5 mm-long Si₃N₄-LiTaO₃ (LTOD) IQ modulator. The waveguides are highlighted in red for better visibility. **b** Schematics of the setup used for the coherent communications experiment. AWG Arbitrary-waveform generator, Amp. RF amplifier, Tx-DSP Transmitter-digital signal processing (off-line), ECL External cavity laser, FPC Fiber polarization controller, EDFA Erbium-doped fiber amplifier, BPF Bandpass filter, TO-PS Thermo-optic phase shifter, ECL External-cavity laser, 90° OH 90° optical hybrid, BPD Balanced photodiode, RTO High-speed real-time oscilloscope (256 GSa/s, 105 GHz), Rx-DSP Receiver-digital signal processing (offline). **c** Measured pre-forward error correction bit error ratio (BER) versus symbol rate for QPSK (brown line) and 16-QAM (dark blue line) signals with horizontal dashed lines indicating the thresholds for soft-decision forward error correction having 15% and 25% coding overhead (15% SD-FEC, 25% SD-FEC)

and for hard-decision FEC with 7% coding overhead (7% HD-FEC). The gray dashed line at the bottom indicates the detection limit, below which our experiment cannot reliably measure BER due to the limited length of the recorded signals (2²³ samples). This limit corresponds to 13 detected bit errors at a symbol rate of 160 GBd with QPSK signals during the recording period. The resulting BER falls within the 99% confidence interval, ranging from half to twice the measured BER[65]. The constellation diagrams of three selected data points, and are shown in (**e–g**). **d** Achievable information rate (AIR, dashed lines) and corresponding net data rate (NDR, solid lines) versus symbol rate for QPSK (brown lines) and 16-QAM (dark blue lines) signals. Constellation diagrams for QPSK signals at symbol rates of 160 GBd (**e**) and 204 GBd (**f**), and for the 16-QAM signals at a symbol rate of 176 GBd (**g**). The corresponding data points are marked by red dashed circles in (**c**).

continuous-wave local oscillator provided by a second ECL to down-convert the optical signal to four electrical baseband signals. Subsequent balanced photodiodes (BPD, Fraunhofer HHI) then detect the resulting waveforms. Finally, as clarified in Supplementary Note 7, the offline receiver DSP recovers and evaluates the signals.

Figure 4c shows the measured pre-FEC BER along with the BER thresholds for different FEC schemes indicated by dashed horizontal lines. Figure 4d provides the corresponding AIR and NDR based on the same calculation used for the IMDD results. Additionally, Supplementary Fig. 9b, c provide the NGMI and error vector magnitude, respectively. For QPSK signals, we obtained reliable BER values only for our measurements at symbol rates of 176 GBd and higher, whereas the measurements at 144 GBd and 160 GBd led to BER values below $1.24 \times 10^{-6}$, which corresponds to the detection limit for the length of

the recorded waveforms. As shown in Fig. 4e, the constellation diagram at 160 GBd features a constellation signal-to-noise ratio (CSNR) of 15.8 dB, which would correspond to an estimated BER of $3.5 \times 10^{-10}$ [48], Eq. 2.18. At the highest QPSK symbol rate of 204 GBd, we measure a BER of $7 \times 10^{-3}$, which is still below the BER threshold for SD-FEC schemes with 25% overhead. In this case, the CSNR amounts to 8.5 dB as displayed in the corresponding constellation diagram from Fig. 4f. Here, the 75 GHz electrical 3 dB-bandwidth of the AWG[49] mainly limits the performance of this experiment. As indicated in Fig. 4c, the highest symbol rate achieved for 16-QAM signaling amounts to 176 GBd and leads to a measured BER of $3.08 \times 10^{-2}$—still below the BER threshold for SD-FEC codes with 25% overhead. This corresponds to a line rate of 704 Gbit/s. The corresponding NDR of up to 581 Gbit/s validates the feasibility of telecommunication systems based on hybrid Si₃N₄-LiTaO₃ circuits. Figure 4g shows the corresponding constellation

diagram, from which we measure a CNSR of 12.0 dB. For our IMDD demonstration, achieving greater data rates will likely require specialized driver electronics given the $Si_3N_4$-$LiTaO_3$ modulator's >100 GHz bandwidth.

To the best of our knowledge, these experiments demonstrate high symbol-rate communication capabilities that surpass the performance metrics of other $Si_3N_4$-based heterogeneous photonic platforms reported to date[50–52]. Meanwhile, our results exceed the previously reported sub 100 GBd symbol rates achieved in hybrid silicon lithium niobate coherent modulators[53].

## Discussion

Alternative hybrid $Si_3N_4$-$LiTaO_3$ waveguide geometries can further improve some of their performance figures. However, such improvements might come at the expense of other metrics. For instance, relying on a thinner bonded $LiTaO_3$ film can reduce slab losses in the resulting hybrid waveguide, thereby leading to lower optical propagation losses and ring resonators with higher quality factors. However, this narrower film will also reduce the overlap of the propagating mode with the electro-optic material, thus resulting in a lower modulation efficiency. Besides such underlying design tradeoffs, some features might require specific waveguide dimensions that will constrain other figures of merit to fixed ranges. For example, a waveguide hosting optical nonlinearities must often verify specific dispersive relations determined by its cross-section[8,10,11]. As a result, distinct features required of the PIC by specific end-uses will ultimately dictate the geometry and hence the performance of its waveguides.

Our $Si_3N_4$-$LiTaO_3$ platform could potentially benefit from additional functionalities brought by platform extensions. Namely, adapting its bonding process could lead to the integration of III-V active optical materials[54–57] and potentially to a multilayer platform featuring both III-V components[58] and $Si_3N_4$-$LiTaO_3$ hybrid waveguides. This wafer-scale integration approach offers a scalable solution that avoids the risk of cross-contamination inherent to processing III-V materials within a CMOS foundry. Incorporating distinct $Si_3N_4$ and $LiTaO_3$ waveguide layers would also introduce benefits, such as eliminating slab losses in $Si_3N_4$ waveguides and increasing mode overlap with $LiTaO_3$ for enhanced electro-optic coupling strengths. However, these benefits will come at the cost of those attributed to the minimal processing of the $LiTaO_3$ film in our fabrication flow.

To summarize, we introduced a photonic platform relying on wafer-scale bonding of $LiTaO_3$ films on foundry-compatible $Si_3N_4$ PICs. Its fabrication process reaches a 100% bonding yield across the wafer's nine central device fields. Compared to other monolithic electro-optic PIC platforms[31,38,39], $Si_3N_4$-$LiTaO_3$ circumvents specialized $LiTaO_3$ etching by relying on a standardized $Si_3N_4$ high-volume process flow. Resulting circuits preserve low optical losses at telecommunication wavelengths while also accommodating broadband modulators featuring high efficiencies and a flat electro-optic response over modulation rates extending up to 100 GHz. Data transmission experiments confirm the practicality of these features by demonstrating net data rates exceeding 500 Gbit/s. Ease of access to high-volume production of $Si_3N_4$-$LiTaO_3$ PICs reinforces its prospects in field-deployable applications not only limited to high-speed optical communications, but that also include microwave-to-optical transducers[59–61] and fast tunable LiDAR[62,63]. Built-in integration with thick $Si_3N_4$ waveguides also enables a new generation of on-chip systems leveraging both electro-optic and low-loss waveguides ranging from microwave oscillators[23,24], to potential interfaces with Kerr frequency combs[8,9].

**Note:** During the preparation of this manuscript, two similar manuscripts[51,52] were posted on the arXiv pre-print repository, but with distinct fabrication processes and experimental results.

## Data availability

The data and code generated in this study have been deposited in the Zenodo database under accession code [https://doi.org/10.5281/zenodo.17276861][64].

## Code availability

The data and code generated in this study have been deposited in the Zenodo database under accession code [https://doi.org/10.5281/zenodo.17276861][64].

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

## Acknowledgements

The PICs were fabricated in the EPFL Center of MicroNanoTechnology (CMi). The LTOI wafers were fabricated in Shanghai Novel Si Integration Technology (NSIT) and the SIMIT-CAS. This work was supported by funding from the Swiss National Science Foundation under grant agreement No. 216493 (HEROIC) awarded to T.J.K., by the European Union's Horizon Europe research and innovation programme under grant No. 101187515 (ELLIPTIC) awarded to T.J.K. and C.K., by funding from the German Research Foundation via the projects PACE (# 403188360) and GOSPEL (# 403187440) awarded to C.K., by funding by the Shanghai Science and Technology program (24CL2901000) awarded to X.O.

## Author contributions

J.C., C.W., and X.J. fabricated the Si$_3$N$_4$-LiTaO$_3$ PICs. J.C., C.W, and J.Z. measured the losses and EO responses of the PICs. J.S. and S.Z. packaged the modulator for DC bias drift measurements. A.K. measured the frequency response. A.K. and D.D. conceived and performed the data transmission experiments together with C.K. and jointly discussed the results. J.C., A.K., D.D., and H.L. analyzed the data. T.J.K., C.K., X.O., and

H.L. supervised all aspects of the project. H.L. and J.C. prepared the content of the manuscript in assistance with contributions and discussions provided by all the authors.

## Competing interests
C.K. and T.J.K. are co-founders and shareholders of Luxtelligence SA, St. Sulpice, Switzerland, a company engaged in electro-optic modulators based on ferroelectric materials. T.J.K. is co-founder and shareholder of LIGENTEC SA, offering heterogenous integration of LN on silicon nitride PICs. The other authors declare no competing interests.
