## [Transparent Peer Review file · Nature Communications]

Heterogeneously integrated lithium tantalate-on-silicon nitride modulators for high-speed communications

Corresponding Author: Professor Tobias Kippenberg

Version 0:

Reviewer comments:

Reviewer #1

(Remarks to the Author)

The authors showed a heterogeneously integrated LiTaO₃ on silicon nitride modulators. The fabrication process is quite similar as previous works of LiNbO₃ on silicon nitride. However, due to improved photostability, low birefringence and enhanced DC bias stability, this work showed enhanced performance of heterogeneous integrated modulators. Nevertheless, some measurements, which could highlight the enhanced performance of heterogeneous integrated modulators, is missing. Wafer level analysis is also missing for this wafer level heterogeneous integration. The authors should add this analysis before this paper could be accepted in Nature Communication. Please see me comments as following:

Main text:

1. In line 48, the authors highlight travelling-wave optical parametric amplifier in 2022 and 2024, but didn't cite the first work from Chalmers in Science advances 2021.
2. Could authors add wafer mapping of the thickness of LiTaO₃ thin film and estimated silicon nitride waveguide height due to the nonuniformity of etching and CMP process? Thickness variation of SiO₂ interlayer? Its impact on modulator?
3. The authors highlight LiTaO₃ superior feature of enhanced DC bias stability compared to LiNbO₃. Could authors add DC bias stability measurement of this heterogeneously integrated LiTaO₃ on Si₃N₄ modulator? Does heterogeneous integration maintain superior feature of enhanced DC bias stability?
4. Heterogeneous integration in Wafer level is crucial for massive production. The authors showed high yield wafer bonding of LiTaO₃ thin film on processed Si₃N₄ wafer through a camera image. However, the authors didn't provide any wafer level analysis of the performance of heterogeneously integrated LiTaO₃ on Si₃N₄ modulator. Could authors provide more detailed analysis? Probably provide 5 measurements regarding a fixed design from 5 different shots.

Supplementary Information

1. Figure S1 has an error, in the process step of DUV lithography, the mask should be resist instead of silicon nitride.
2. The authors' previous work typically use LPCVD TEOS, HTO, LTO. Could authors explain why ICPCVD SiO₂ is used as interlayer in this work?
3. The authors use a inversed LiTaO₃ taper with tip width of 500nm to form a low-loss transition from Si₃N₄ waveguide to Si₃N₄/LiTaO₃ hybrid waveguide. However, the large LiTaO₃ tip introduce insertion loss of near 0.28 dB. Why not use a even narrower tip to further reduce the insertion loss? Could authors comment the introduced return loss, and its potential impact on optical communication?

Reviewer #2

(Remarks to the Author)

The manuscript positions itself as the first wafer-scale LT-on-Si₃N₄ platform achieving ~100 GHz electro-optic bandwidth and high-symbol-rate IM/DD and coherent communications on-chip. This novelty claim is broadly convincing, as earlier LT efforts emphasized either monolithic LTOI photonics or discrete thin-film LT modulators without co-integration on Si₃N₄, while prior heterogeneous integration on Si₃N₄ used LiNbO₃ (LN). By transposing the proven LN-on-Si₃N₄ heterogeneous template to LT and then validating with both IM/DD and coherent formats the manuscript advances the material-platform combination in a way that is more than cosmetic. That said, the narrative would benefit from a concise related-work

comparison in a paragraph or table format (bandwidth, symbol rate, efficiency, routing loss) to precisely delimit what is new and what is inherited.

The bonding and thinning flow is described in a level of detail suitable for reproduction in a capable fab, including hydrophilic pre-bonding and a moderate-temperature anneal. Avoiding aggressive plasma exposure of LT during transfer is methodologically sound. The wafer-level photograph supports the 'wafer-scale' claim qualitatively, and the text asserts complete bonding across nine central fields. However, for genuine manufacturing credibility the paper should substantiate this with wafer maps and statistics, showing device-to-device distributions (V_{π} , $V_{\pi L}$, bandwidth, insertion loss) and a quantitative definition of 'bonding yield' in terms of devices meeting specific specs. This addition would convincingly elevate the scalability argument.

Loss extraction via ring-resonator linewidths is appropriate for a hybrid stack. The reported ~ 14.2 dB/m is reasonable for a first-generation LT-on- Si_3N_4 hybrid mode but, in absolute terms, is higher than state-of-the-art bare Si_3N_4 routing and also somewhat above the best published heterogeneous LN-on- Si_3N_4 values. Because the abstract uses the phrase 'ultralow loss,' the manuscript should reframe the language and present the number contextually, explicitly distinguishing hybrid-mode loss from bare-core Si_3N_4 , summarizing the measurement method and uncertainty, and clarifying whether the figure reflects the hybrid optical mode or the Si_3N_4 core. Providing measured mode-converter insertion loss (rather than simulation only) would further clarify where loss accrues in large-scale circuits. In line "79" the statement – "ultralow optical propagation losses of $O(\text{dB/m})$ " is also misleading as the material is not lossless, please find a way to mitigate the statement. The inclusion of a measured $V_{\pi L} \approx 4.08$ V-cm is a strength, as it enables cross-platform comparisons without ambiguity. The text also references simulations that predict a somewhat larger $V_{\pi L}$ for nominal geometry, which is encouraging but calls for reconciliation. A discussion aligning measured and simulated results, stating the electro-optic coefficients used, optical and microwave mode overlaps, exact electrode geometry, and layer thicknesses, would resolve possible confusion and reinforce confidence in the reported efficiency.

Broadband S-parameter data up to ~ 110 GHz, including $|S_{11}|$ below -15 dB, indicate careful impedance control and low reflection at the device ports. The small-signal $|S_{21}|$ roll-off appears consistent with ohmic attenuation ($\propto \sqrt{f}$) and modest velocity mismatch. For completeness and reproducibility, the paper should aggregate RF attenuation in dB/cm versus frequency with line length, calibration plane, and termination explicitly stated, and add S_{22} if available. Because recent LT work has highlighted the conductivity advantages of silver electrodes above 100 GHz, a short discussion of metal choice (Au vs Ag) and anticipated performance–reliability trade-offs would be valuable for readers planning to push to even higher bandwidths.

The communications results are a major strength of the manuscript. Demonstrations of PAM-4 in the ~ 192 GBd range yielding ~ 333 Gb/s net after FEC, and coherent 16-QAM around 176 GBd yielding ~ 581 Gb/s net, establish that the devices function beyond small-signal curiosity. The setups are described clearly and the constraints of the electrical instrumentation are acknowledged (e.g., AWG bandwidth limiting the 204-GBd QPSK case). To fully enable external benchmarking, the manuscript should report per-format pre-FEC BER and EVM, the precise FEC overheads and thresholds, estimated or measured energy-per-bit based on the drive swing at the device pads, and a compact link budget that separates fiber-to-chip coupling, on-chip routing, modulator section, and any amplification used.

The introduction fairly motivates LT over LN with lower birefringence and improved photostability and DC-bias stability. However, these advantages are not yet quantified on the presented platform. Including time-resolved bias drift under realistic optical powers and over a relevant temperature range would turn a qualitative motivation into a platform-level evidence point. Even a modest A/B comparison versus a representative LN-on- Si_3N_4 control device, if available, would strengthen the case materially.

The text describes fiber-to-chip coupling on the order of several dB per facet and indicates the need for amplification to meet receiver power targets in some experiments, which is entirely plausible for early-stage devices. Nevertheless, a clear separation of external-coupling loss from intrinsic on-chip insertion loss of the MZM and IQM would meaningfully assist practitioners. Reporting measured mode-converter loss, on-chip device IL, and facet coupling with uncertainties would also help reconcile the communications-experiment power budgets with the passive-loss measurements.

Overall methods transparency is good and likely, nevertheless the paper would still benefit from explicit uncertainty estimates for $V_{\pi L}$, loss extractions, and bandwidth fitting; precise VNA calibration details and de-embedding steps; and, where applicable, references to public PDK-like parameters (layer thicknesses, refractive indices at test wavelengths). Adding micrographs or cross-sections of critical transitions (e.g., vertical adiabatic converters) would also round out the reproducibility story.

Monolithic LTOI platforms have reported low propagation loss and strong efficiency at ~ 40 -GHz-class bandwidths; discrete thin-film LT modulators have reached beyond 100 GHz; heterogeneous LN-on- Si_3N_4 has already proven the architectural value of marrying a strong Pockels material to ultra-low-loss routing. This manuscript's unique contribution is to combine the LT material with Si_3N_4 at wafer scale and then validate with both IM/DD and coherent communications at symbol rates competitive with the best contemporary demonstrations. Given the sustained interest in coherent transceivers and RF-photonics, the practical impact is credible, particularly if the authors add the benchmarking details outlined above and as such the manuscript could be accepted for publications after the requested revisions have been added.

Reviewer #3

(Remarks to the Author)

This is a very good paper which addresses one of the current largest bottlenecks associated with high speed data transmission. Silicon and lithium niobate based optical modulators are areas of active research and the competing technologies; Single lane speeds are limited by modulator speeds and therefore, companies and researchers are intensely studying ways to achieve better on-chip modulators. In this paper authored by Cai et. al, they introduce lithium tantalate based modulators which are integrated with ultra-low loss silicon nitride waveguide devices. I recommend publishing this paper and have some suggestions for the authors:

- 1) The reported V_{π} is 6V. Can the authors comment on how this compares with other state of the art modulators? A commonly used figure of merit is $V_{\pi}L$. A table which compares this figure of merit, bandwidth and other commonly used performance metrics with other modulators would be great for readers.
- 2) The authors demonstrate the modulator platform with both IMDD and coherent data transmission. Can they comment on whether the highest rates demonstrated were limited by the equipment or the modulator platform itself?
- 3) Because of the lack of an easily available direct bandgap material in CMOS, III-V continues to be relevant. In the discussion section, the authors discuss III-V active materials and their potential integration with the lithium tantalate-Si₃N₄ platform in the future through bonding. Can the authors comment on the difficulties that may be associated with the adoption of this in CMOS foundries.
- 4) In a similar vein, lithium tantalate is not currently widely used in CMOS foundries. Some discussion about what could accelerate adoption in foundries would be useful.

Reviewer #4

(Remarks to the Author)

The authors achieved a wafer-scale heterogeneous bonding of LiTaO₃ onto Si₃N₄ PIC platform. Compared to monolithic electro-optic photonics platforms, this approach leverages the low-loss Si₃N₄ waveguides while simultaneously exploiting the electro-optic properties of LiTaO₃. The work demonstrates EO modulation with a V_{π} of 6 V and bandwidths up to 100 GHz, as well as both IMDD and coherent signal transmission, showing performance comparable to EO modulators realized on other PIC platforms. The manuscript is suitable for publication, but the authors should address the following points before acceptance:

1. Compared to state-of-the-art EO modulators, this work exhibits relatively low modulation efficiency, with a reported $V_{\pi}L$ of 4.08 V-cm. For the MZM used in coherent data transmission, the electrode length is up to 13.5 mm. What is the underlying reason for this low efficiency? Is it due to limited optical mode overlap or insufficient electric-field overlap with the active LiTaO₃ film?
2. Was the EO MZM measured under a DC bias? If so, was any bias drift observed, and how severe or minor was it compared to that in lithium niobate devices?
3. In line 251, some works in other heterogeneous platforms are cited (mainly LiNbO₃ or LiTaO₃). However, there have also been similar demonstrations using other EO materials, such as barium titanate or SiN PIC platforms, which provide comparable integration advantages but with significantly larger Pockels coefficients. These works should also be discussed in comparison. In addition, including other types of EO modulators and summarizing performance metrics (both monolithic and heterogeneous) in a comparison table would make the advantages of this heterogeneous Si₃N₄-LiTaO₃ platform over other photonic platforms clearer.
4. The advantages of LiTaO₃ over lithium niobate in this context of modulators were not really very clear to me – lower birefringence and reduced photorefractive effects were mentioned, but how significant is the advantage (especially if we were to compare LN-with-nitride and LT-with-nitride side by side)? I suppose the “fast tunable LIDAR” application makes sense, but a more quantitative discussion would strengthen the motivation.

Reviewer #5

(Remarks to the Author)

Version 1:

Reviewer comments:

Reviewer #1

(Remarks to the Author)

The authors replied my comments and revised the manuscript properly. The manuscript can be published in Nature Communication.

In supplementary figure 5.c, the unit of x-axis is incorrect. it should be MHz.

Reviewer #2

(Remarks to the Author)

In the updated manuscript and supplementary information, the authors have addressed my concerns. One last comment

would be to include the device insertion losses in the new Supplementary Table 3.(Performance metrics of high-speed EO modulators).

In my opinion, the innovation and engineering values of the work are now good for the publication.

Reviewer #3

(Remarks to the Author)

The authors have addressed my comments. I recommend publishing the manuscript.

Reviewer #4

(Remarks to the Author)

I'm satisfied by the replies to the points I raised in my earlier review.

A point-by-point response to the reviews' comments of the manuscript "Heterogeneously integrated lithium tantalate-on-silicon nitride modulators for high-speed communications"

We would like to express our sincere gratitude to all the reviewers for their detailed review and insightful suggestions regarding the manuscript. Here, we provide point-by-point answers to the reviewers' comments. For clarity, the original text of the comments is shown in black, the replies are in blue, and the actions taken are highlighted in red and corresponding modifications in the manuscript are underlined.

Reviewer #1 (Remarks to the Author):

The authors showed a heterogeneously integrated LiTaO₃ on silicon nitride modulators. The fabrication process is quite similar as previous works of LiNbO₃ on silicon nitride. However, due to improved photostability, low birefringence and enhanced DC bias stability, this work showed enhanced performance of heterogeneous integrated modulators. Nevertheless, some measurements, which could highlight the enhanced performance of heterogeneous integrated modulators, is missing. Wafer level analysis is also missing for this wafer level heterogeneous integration. The authors should add this analysis before this paper could be accepted in Nature Communication. Please see me comments as following:

Reply: We are grateful for the reviewer's recognition of the enhanced performance our work demonstrates—notably the advancements on broadband electro-optic modulation and record-high data transmission rate. We also thank the reviewer for the constructive comments on the wafer-scale measurements to further highlight the capabilities of our platform and corresponding heterogeneous integration technique. We have addressed these points carefully in our response below.

Main text:

1. In line 48, the authors highlight travelling-wave optical parametric amplifier in 2022 and 2024, but didn't cite the first work from Chalmers in Science advances 2021.

Reply: We thank the reviewer for the careful literature review and mentioning the missing citation on integrated optical parametric amplifier (OPA). In our revised version, we have now added the citations of the first integrated OPA work from Ye, Z. et al. (Science advances 7.38 (2021): eabi8150.).

Action taken: To highlight the capacity of silicon nitride waveguide on on-chip OPA, we have updated our reference as below:

New citations in main text (Page 1):

[16] Z. Ye, P. Zhao, K. Twayana, M. Karlsson, V. Torres Company, and P. A. Andrekson, Overcoming the quantum limit of optical amplification in monolithic waveguides, Science Advances 7, eabi8150 (2021).

2. Could authors add wafer mapping of the thickness of LiTaO₃ thin film and estimated silicon nitride waveguide height due to the nonuniformity of etching and CMP process? Thickness variation of SiO₂ interlayer? Its impact on modulator?

Reply: We thank the reviewer for raising this concern on wafer-scale uniformity. For the LiTaO₃ thin film, the LiTaO₃-on-insulator (LTOI) wafers are first produced via ion slicing (Smart-Cut) followed by bonding to a silicon wafer covered with wet oxide. We then use this wafer to bond the LiTaO₃ layer onto the Damascene photonic Si₃N₄ wafer. This technique preserves the original thickness variations of the bonded thin film, which has a nonuniformity of less than 5% (Supplementary figure 4(b)). Instead of using a LiTaO₃ film from a LTOI wafer, we could directly bond an ion-sliced LiTaO₃ film from a bulk wafer to our patterned silicon nitride wafer as performed for lithium niobate in [Laser Photonics Rev. e00138 (2025)]. Such an approach would circumvent any film degradation experienced by the LiTaO₃ film in the preparation of the LTOI wafer used for this work.

For the Si_3N_4 layer, as seen in what is now Supplementary figure 4(a) of our revised SI, its variation typically lies on the scale of several tens of nanometers and arises primarily from the CMP process. Specifically, the process is sensitive to the polishing stress distribution and surface relief induced by small filler patterns. As emphasized in Supplementary Figure 4(c), these thickness variations can affect the modulator's half-voltage (V_{π}) and optical loss, albeit to a negligible extent.

For the interlayer oxide, its thickness lies around 50 nm and realistically only varies by a few nanometers. Such small variations have a negligible impact on both the optical effective index and the electro-optic mode overlap and thus does not affect the modulator's performance.

Action taken: In the Supplementary Information (SI), we have put our measured wafer mapping data of the LiTaO_3 thin film and Si_3N_4 waveguide height using an optical profilometer (FilMetrics F54-XY-200). We have also simulated the modulator's optical loss and V_{π} for different Si_3N_4 waveguide heights. We then discuss the non-uniformity sources for each layer and analyze their impacts on the modulator's performance.

Addition to the main text (**Page 3**):

Supplementary Note 3 provides further details on wafer-scale yields, which include film thickness homogeneity and device performance analyses.

Addition to the SI (**Supplementary Section 3**):

Supplementary Figure 4. Analysis of thickness variations and their impact on device performance. Thickness maps of the (a) silicon nitride layer and (b) lithium tantalate thin film. The maps are interpolated from measurements at the nine central stepper fields (marked as black dots). Simulated modulation efficiency and optical loss as a function of (c) the silicon nitride waveguide height and (d) the lithium tantalate thin film thickness. The shaded regions indicate the measured range of film thickness on the wafers.

To quantify the wafer-scale process yield of the Si_3N_4 - LiTaO_3 platform, Supplementary Figures 4(a,b) show the measured thickness maps of the Si_3N_4 waveguide (post-

Damascene process) and thin film LiTaO₃ (post-bonding) layers, respectively. The raw data, measured at the center of the nine stepper fields defined in Supplementary Figure 5(a) are (532 nm, 539 nm, 511 nm; 559 nm, 571 nm, 531 nm; 535 nm, 543 nm, 505 nm) for the Si₃N₄ layer and (306 nm, 300 nm, 298 nm; 293 nm, 295 nm, 289 nm; 294 nm, 298 nm, 299 nm) for the LiTaO₃ layer. The Si₃N₄ thickness features height variations introduced by the surface polishing step of the Damascene process. Additional CMP on the interlayer used for bonding can further modify the absolute Si₃N₄ thickness across the wafer. In contrast, the bonded LiTaO₃ thin film exhibits much higher uniformity, with total variations near 17 nm.

Based on the simulated results in Supplementary Figure 4(c,d), increasing the waveguide height within the measured range of 500 to 560 nm only slightly decreases optical loss due to stronger mode confinement in the Si₃N₄ core. This reduction comes at the cost of an increased V_π from 4.55 to 4.85 V. As for the LiTaO₃ thickness, a tens-of-nanometer variation significantly alters V_π, while a thicker LiTaO₃ thin film increases metal-induced absorption loss. Therefore, we identify the uniformity of the LiTaO₃ layer as the most critical parameter for achieving consistent device performance. The excellent uniformity provided by the ion-slicing and wafer-bonding techniques is therefore crucial for this platform. To further improve wafer-level device consistency, using a subtractive process to fabricate the Si₃N₄ waveguide [7] can mitigate dimensional changes that arise from the polishing and thermal reflow fabrication steps. However, achieving a flat surface required for high-yield bonding is considerably more complex with this subtractive method compared to the Damascene process.

3. The authors highlight LiTaO₃ superior feature of enhanced DC bias stability compared to LiNbO₃. Could authors add DC bias stability measurement of this heterogeneously integrated LiTaO₃ on Si₃N₄ modulator? Does heterogeneous integration maintain superior feature of enhanced DC bias stability?

Reply: Thanks for the reviewer's suggestion. Although prior work (C. Wang, Nature 629, 784 (2024)) had shown the superior modulation stability inherited from LiTaO₃, adding a DC bias drift measurement is still reasonable to demonstrate our platform's advantage in preserving this intrinsic material property. Here, the measurements are carried out through a homemade photonic packaging process, thereby removing the effects of chip-to-fiber alignment drift while coupling the modulator's output to a fiber mounted on a flexure stage. By setting the packaged modulator at its working point (half V_π for the balanced MZM), the measured DC bias drift exhibits a low power shift of less than 0.5 dB over one hour, thus highlighting its superior performance compared to the current LiTaO₃-based performance benchmark [Optica 11, 1614-1620 (2024), Optics Express 32, 44115 (2024)]. We believe that this progress is largely due to the stable fiber-to-chip coupling and lower defect density of the unetched LiTaO₃ layer, which provides a baseline for future applications.

Action taken: We have added a new paragraph to the main text highlighting the DC bias drift measurement and updated Figure 2 to highlight this aspect of our platform. We have also added a new section to the SI, including the image of the packaged LiTaO₃ -Si₃N₄ modulator and details about our packaging method.

Modification to the main text (**Figure 2 and Page 5**):

Figure 2. Hybrid Si_3N_4 - LiTaO_3 electro-optic modulators. (a) Optical micrograph of a fabricated 6.8 mm-long hybrid modulator. Insets: optical micrographs of the modulator's underlying components, which include Y-splitters, a coplanar waveguide electrode, and tapered transitions between Si_3N_4 waveguides and Si_3N_4 - LiTaO_3 waveguides with a simulated 0.28 dB insertion loss. (b) False-colored scanning electron microscopy (SEM) image of the cross-section of a manufactured Si_3N_4 - LiTaO_3 modulator. (c) Normalized transmission of a 6.8 mm-long push-pull MZM versus applied voltage. The inset diagram shows the high-speed electrode geometry with a 23 μm signal width and a 6 μm electrode gap. (d) Measured electro-optic response (electro-optic S_{21}) and microwave return loss (electrical reflection S_{11}), revealing a high 3-dB bandwidth near 100 GHz. (e) Output optical intensity of the modulator while biased at its working voltage, i.e. its quadrature point, over one hour.

Monitoring the modulator's optical output power while set to its quadrature point quantified its stability under a DC bias. As outlined in Supplementary Note 6, a dedicated photonic package mitigated the influence of drifting chip-to-fiber coupling efficiency in this measurement. As shown in Fig. 2(e), the Si_3N_4 - LiTaO_3 MZM exhibits low power shifts of less than 0.5 dB over a one-hour period, thus outperforming results reported in prior work on monolithic lithium niobate and lithium tantalate PIC platforms [38, 40, 41]. Minimal degradation of the unetched LiTaO_3 film, where defect-driven photorefractive effects typically give rise to drift [42], likely contributes to the modulator's enhanced DC bias stability.

Addition to the SI (Supplementary Section 6):

6. Photonic package for DC-bias drift measurements

Supplementary Figure 8. Photonic package for DC electro-optic stability measurements. Image of the packaged Si_3N_4 - LiTaO_3 modulator.

The DC bias drift of a modulator consists of the temporal variations in its optical output power while biased at a constant voltage (e.g., the quadrature point). These fluctuations typically arise from mechanisms such as charge relaxation and the photorefractive effect [14, 15]. Though the DC-stability of monolithic LiTaO_3 modulators was demonstrated in previous work [16], a similar study is still essential to verify whether this feature also applies to hybrid Si_3N_4 - LiTaO_3 MZMs. To remove the influence of varying chip-to-fiber coupling efficiency, a robust and custom photonic package was engineered for this experiment. As shown in Supplementary Figure 8, the chip was first manually mounted onto a copper sub-mount with a UV-cure adhesive and then electrically connected with wire bonding (F&S Bondtec 56i). This step was followed by active alignment of the input/output fibers that were precisely maneuvered using multi-axis fixture, while the optical power was monitored with a photodiode. Once the coupling efficiency was optimized, the fibers were permanently fixed in place at the chip facets using high-precision epoxies.

4. Heterogeneous integration in Wafer level is crucial for massive production. The authors showed high yield wafer bonding of LiTaO_3 thin film on processed Si_3N_4 wafer through a camera image. However, the authors didn't provide any wafer level analysis of the performance of heterogeneously integrated LiTaO_3 on Si_3N_4 modulator. Could authors provide more detailed analysis? Probably provide 5 measurements regarding a fixed design from 5 different shots.

Reply: We thank the reviewer for this constructive comment. Demonstrating wafer-level performance is essential to validate the potential of our platform for volume manufacturing. We have selected five fields across the main area of the wafer and conducted a series of measurements on nominally identical devices including ring resonators, long, and short modulators. Besides higher levels of optical losses in one field (F7), these properties are consistent across the wafer.

Action taken: We have added a new section to the SI, which presents the detailed wafer-level analysis of key performance metrics (high-frequency EE response, frequency microwave loss, optical loss and half-wave voltage) from five distinct fields. The attached illustration summarizes these important uniformity results.

Addition to the SI (**Supplementary Section 3**):

Supplementary Figure 5. Wafer-scale characterization of EO modulators and microring resonators. (a) Schematic showing the five fields selected for wafer-level testing (left panel) and the selected chips for inter-field device comparison (right panel). (b) Measured microwave electrical-to-electrical S21 response for 6.8 mm long and 16 mm long CPW electrodes, showing high uniformity across the five fields. (c) Intrinsic linewidth, $\kappa_0/2\pi$, histogram for microrings from the selected fields. All rings share the same design featuring a 112 GHz free spectral range. (d,e) Measured modulation efficiency and RF loss slope for three 6.8 mm MZMs within C4. The modulators share the same signal-ground gap yet have different signal electrode widths. Missing data points are due to device damage during handling or packaging.

As depicted in Supplementary Figure 5(a), device performance uniformity across the wafer is characterized by choosing five representative fields for testing as well as selecting three chips within a given field for evaluating inter-field device variations. From the obtained microwave transmission displayed in Supplementary Figure 5(b), the 3 dB EO bandwidth can be approximately calculated from the 6 dB electrical bandwidth when only considering the effects of microwave loss and assuming impedance matching and velocity matching [8, 9]. The electrical S21 curves for both the 6.8 mm and 16 mm long devices are tightly grouped, indicating excellent consistency. Correspondingly, the estimated 3 dB EO bandwidth exceeds 67 GHz for the 6.8 mm modulators and is approximately 20 GHz for the 16 mm modulators across all tested fields. Optical loss uniformity was evaluated by measuring the intrinsic linewidth $\kappa_0/2\pi$ of microring resonators across the same five fields. Supplementary Figure 5(c) provides intrinsic linewidth histograms of these rings' resonances over a considered frequency range of 184-200 THz. The microrings are located on C5 and

share the same design featuring a 112 GHz free-spectral range. Relevant statistics presented as (Field number, Mean value, Median value, Max value) are listed as follows: (F1, 66.8 MHz, 58.6 MHz, 45 MHz), (F3, 69.1 MHz, 57.7 MHz, 55 MHz), (F5, 77.8 MHz, 71.7 MHz, 55 MHz), (F7, 164 MHz, 187 MHz, 50 MHz), (F9, 85.6 MHz, 57.1 MHz, 55 MHz). Inter- and intra-field uniformity is highlighted in Supplementary Figure 7(d,e). The modulation efficiency and RF loss for three 6.8 mm long MZMs, with the same electrode gap and different signal widths (marked as Devices 101-103) in C4, demonstrate comparable performance to the reported device in the main text, with minor variations attributed to fabrication tolerances.

Supplementary Information

1. Figure S1 has an error, in the process step of DUV lithography, the mask should be resist instead of silicon nitride.

Reply: We thank the reviewer for catching this mistake in the SI. The layer should be marked as red to represent the 'Resist' for patterning the silicon nitride waveguide.

Action taken: We have modified Supplementary Figure 1 as shown below:

2. The authors' previous work typically use LPCVD TEOS, HTO, LTO. Could authors explain why ICPCVD SiO_2 is used as interlayer in this work?

Reply: We sincerely thank the reviewer for raising this question, which allows us to correct a significant typo in the manuscript. The typo is that the interlayer is indeed deposited using LPCVD rather than ICPCVD. The mistake is because some of our more recent works have utilized an ICPCVD process with a SiCl_4 precursor to deposit a high-density and hydrogen-free cladding, which is an efficient deposition method as detailed in [arXiv preprint arXiv:2312.07203 (2023)]. However, for the specific requirements of the high-speed modulator, ICPCVD is not a suitable choice owing to the involved high-power oxygen plasma (normally exceeding 2000 W). This kind of plasma could introduce defects at the Si/SiO₂ interface, thereby leading to significant parasitic conduction losses at microwave frequencies. Therefore, we chose the established method of LPCVD followed by annealing to ensure the lowest RF loss.

Action taken: We have corrected the description of the interlayer deposition method in the SI.

Revision to the SI (**Supplementary Section 1**):

A 1 μm thick oxide interlayer is deposited by LPCVD, after which an additional 1200 $^\circ\text{C}$ annealing is conducted to eliminate absorption losses associated with hydrogen impurities. Finally, a second CMP step is adopted to polish the interlayer to a thickness of 50 nm so as to remove the residual topography due to the Damascene process, as well as ensuring sub-nanometer surface roughness for subsequent wafer bonding of lithium tantalate on Damascene silicon nitride wafers.

3. The authors use a inverted LiTaO_3 taper with tip width of 500nm to form a low-loss transition from Si_3N_4 waveguide to $\text{Si}_3\text{N}_4/\text{LiTaO}_3$ hybrid waveguide. However, the large LiTaO_3 tip

introduce insertion loss of near 0.28 dB. Why not use a even narrower tip to further reduce the insertion loss? Could authors comment the introduced return loss, and its potential impact on optical communication?

Reply: We would like to thank the reviewer for these insightful questions. Regrettably, our current design does not include test structures (e.g., cascaded tapers) required to accurately de-embed the loss profile of a single taper transition, but we would like to offer a detailed discussion on this topic.

Regarding the taper tip width and its effect on insertion loss, we fully agree that a narrower tip width would further reduce the mode transition loss, which is proven by our new simulation using different tip widths (Supplementary Note 4). The 500 nm tip width in this work is a direct consequence of the resolution limit of our maskless lithography tool (Heidelberg Instruments MLA150). This could be improved by migrating this lithography step to a higher-resolution technique, such as DUV and electron-beam lithography, whose resolution can readily achieve 150 nm or less.

Furthermore, we acknowledge that any non-ideal mode transition can introduce a return loss. However, the back reflection from a single taper is sufficiently low so as not to create a Fabry-Pérot cavity effect between input and output mode transitions, thus we are confident that this was not a dominant factor impairing our high-speed EO modulation. The quality of our experimental communications results presented in the manuscript provides strong evidence for the above conclusion.

Action taken: We have updated the main text to clarify that the current taper geometry is limited by our lithography resolution and have added a discussion on the potential for performance improvement using DUV lithography. In the SI, we have updated the simulation to consider tip widths from 200 nm to 500 nm. The results show the potential improvements of using a narrower transition taper.

Revision to the SI (**Supplementary Section 4**):

As shown in Supplementary Figure 6(c), the 500 nm minimal width of the taper primarily limits transmission. The patterning of the LiTaO₃ layer relied on a mask-less lithography method. As shown in the scanning electron micrograph of Supplementary Figure 6(d), this type of lithography produces coarser features resulting in greater insertion losses. The simulated data provided in Supplementary Figure 6(b) suggest that decreasing this width down to 200 nm can decrease the taper's insertion loss down to 0.08 dB. As previously demonstrated for heterogeneously integrated LiNbO₃-Si₃N₄ circuits, DUV optical lithography can reach these widths and insertion loss figures.

Supplementary Figure 6 Simulated transmission of silicon nitride-lithium tantalate waveguide transitions. (a) Schematic diagram and corresponding FDTD simulations of the adiabatic coupling from the Si_3N_4 waveguide to the hybrid Si_3N_4 - LiTaO_3 waveguide. (b) Simulated transmission spectrum of the waveguide transition for various minimum LiTaO_3 taper widths. (c) Simulated electric field distribution of the adiabatic taper for a 500 nm minimum LiTaO_3 taper width. (d) Top-view scanning electron micrograph of the waveguide transition where the patterned LiTaO_3 layer features blue false colors.

Reviewer #2 (Remarks to the Author):

The manuscript positions itself as the first waferscale -LT--on--Si₃N₄ platform achieving ~100 GHz electro-optic bandwidth and high -symbol rate IM/DD and coherent communications -onchip. This novelty claim is broadly convincing, as earlier LT efforts emphasized either monolithic LTOI photonics or discrete -thin-film- LT modulators without co-integration on Si₃N₄, while prior heterogeneous integration on Si₃N₄ used LiNbO₃ (LN). By transposing the proven LN--on--Si₃N₄ heterogeneous template to LT and then validating with both IM/DD and coherent formats the manuscript advances the material–platform combination in a way that is more than cosmetic. That said, the narrative would benefit from a concise related w-ork comparison in a paragraph or table format (bandwidth, symbol rate, efficiency, routing loss) to precisely delimit what is new and what is inherited.

Reply: We sincerely appreciate the reviewer’s acknowledgements of our efforts in advancing heterogenous photonic platforms exploiting multi-material capabilities and pushing towards practical telecommunication applications. Following the reviewer’s constructive suggestion, we completely agree that a comprehensive comparison to related works helps readers precisely understand the advances made by our platform . As such, we have added a table in the SI to compare our work with other state-of-the-art platforms.

Action taken: In accordance with the reviewer’s suggestion, we have added a comparison table in the SI (**Supplementary Note 9**). This table lists the key performance metrics (e.g. Bandwidth, V_πL, Data rate, BER) of current integrated EO modulator technologies, which includes monolithic thin film (LNOI, LTOI, BTOOI) and recent heterogeneous works (BTO-on-Si₃N₄, LiNbO₃-on-Si₃N₄ and LiTaO₃-Si₃N₄).

Addition to SI (**Supplementary Section 9**):

8. Comparison to the state-of-the-art

To benchmark our device against the state-of-the-art, we summarize the performance metrics of leading EO modulator technologies in Supplementary Table 3. These technologies include monolithic platforms such as LiNbO₃-on-insulator (LNOI), LiTaO₃-on-insulator (LTOI) and BaTiO₃-on-insulator (BTOOI) as well as heterogeneous photonic platforms based on BaTiO₃-Si₃N₄, LiNbO₃-Si₃N₄ and LiTaO₃-Si₃N₄. Our devices exhibit a compelling performance profile for both single MZM and IQ modulator architectures by achieving superior data throughput while maintaining competitive modulation efficiency.

Supplementary Table 3. Performance metrics of high-speed EO modulators.

Ref.	Platform	Type	Bandwidth [GHz]	V _π L [V·cm]	Format	Line rate [Gbit/s]	BER
29	LNOI	single MZM	> 45	2.8	PAM 4	140	2.10 × 10 ⁻⁵
	LNOI	single MZM	> 45	2.8	PAM 8	210	1.50 × 10 ⁻²
	LNOI	single MZM	~ 100	2.2	NA	NA	NA
30	LNOI	single MZM	~ 170	3.3	PAM 4	200	5.50 × 10 ⁻³
	LNOI	single MZM	~ 170	3.3	PAM 8	240	1.10 × 10 ⁻²
31	LNOI	IQ	48	2.47	QPSK	220	8.63 × 10 ⁻⁶
	LNOI	IQ	48	2.47	16 QAM	320	8.41 × 10 ⁻³
32	LNOI	IQ	110	2.3	DP 16 QAM	1600	NA
	LNOI	IQ	110	2.3	DP 64 QAM	2220	NA

11	LTOI	single MZM	~ 110	2.88	PAM 8	528	3.80×10^{-2}
33	BTOOI	single MZM	NA	2.32	NA	NA	NA
34	BaTiO ₃ -Si ₃ N ₄	single MZM	NA	0.48	PAM 4	212	3.50×10^{-3}
35	LiTaO ₃ -Si ₃ N ₄	single MZM	~ 70	2.3	PAM 4	320	$\sim 1.60 \times 10^{-2}$
36	LiNbO ₃ -Si ₃ N ₄	single MZM	~ 110	3.40	NA	NA	NA
This work	LiTaO₃-Si₃N₄	single MZM	100	4.08	PAM 4	400	3.00×10^{-2}
This work	LiTaO₃-Si₃N₄	IQ	34	4.05	QPSK	408	7.00×10^{-3}
This work	LiTaO₃-Si₃N₄	IQ	34	4.05	16 QAM	704	3.08×10^{-2}

The bonding and thinning flow is described in a level of detail suitable for reproduction in a capable fab, including hydrophilic prebonding and a -moderate temperature anneal. Avoiding aggressive plasma exposure of LT during transfer is methodologically sound. The -wafer-level photograph supports the '-wafer-scale' claim qualitatively, and the text asserts complete bonding across nine central fields. However, for genuine manufacturing credibility the paper should substantiate this with wafer maps and statistics, showing -device--to-device- distributions (V_{π} , $V_{\pi L}$, bandwidth, insertion loss) and a quantitative definition of 'bonding yield' in terms of devices meeting specific specs. This addition would convincingly elevate the scalability argument.

Reply: We would like to thank the reviewer for the positive feedback on our fabrication technique and highlighting the necessity of wafer-level statistics. Concerning the device performance across the whole wafer, we ideally would present measured data from all nine central fields. However, several devices with available designs were already committed to cleaving, packaging and high-speed telecommunication testing in collaborating labs. To provide a representative performance map, we have characterized identical devices from five distributed fields spanning the full bonded area. This analysis, now presented in the SI, provides the device-to-device distributions for key performance metrics (e.g. modulation efficiency, bandwidth, optical loss).

For the 'bonding yield' mentioned by the reviewer, we define it as the percentage of devices that meet a specific set of performance specifications. To quantify this, yielding a complete analysis would require testing a very large population of different devices and even multiple fabrication runs, which is essential for commercial manufacturing but beyond the scope of this initial demonstration. However, to provide more quantitative figures involving the bonded wafers, we now provide the thickness mapping for nine field to further emphasize our wafer-scale integration capabilities. Furthermore, the wafer-scale device measurements mentioned above displays a level of uniformity serving as a strong proxy for the commercial potential and high scalability of our platform.

Action taken: In the SI, we have added a new section titled "Wafer-level characterization" with Supplementary Figure 4 and Supplementary Figure 5, which presents the detailed wafer-level analysis of thickness mapping and key performance metrics for MZMs (high-frequency electric-to-electric (EE) response, frequency microwave loss, optical loss, and half-wave voltage) from five distinct fields.

To avoid repetition, the modification details can be seen in our above response to **Reviewer #1, Comment #2 and Comment #4.**

Loss extraction via ring-resonator linewidths is appropriate for a hybrid stack. The reported ~ 14.2 dB/m is reasonable for a first-generation LT-on-Si₃N₄ hybrid mode but, in absolute terms, is higher than state-of-the-art bare Si₃N₄ routing and also somewhat above the best published heterogeneous LN-on-Si₃N₄ values. Because the abstract uses the phrase ‘ultralow loss,’ the manuscript should reframe the language and present the number contextually, explicitly distinguishing hybrid-mode loss from bare-core Si₃N₄, summarizing the measurement method and uncertainty, and clarifying whether the figure reflects the hybrid optical mode or the Si₃N₄ core. Providing measured mode-converter insertion loss (rather than simulation only) would further clarify where loss accrues in largescale circuits. In line “79” the statement – “ultralow optical propagation losses of O(dB/m)-” is also misleading as the material is not lossless, please find a way to mitigate the statement.

Reply: We thank the reviewer for pointing out the inaccurate statement on optical loss in this manuscript. We would like to clarify that the reported loss value of 14.2 dB/m was extracted solely from the linewidth of the LiTaO₃-on-Si₃N₄ ring resonators, which are specifically designed to probe their losses. Compared to other state-of-the-art photonic integrated circuits, the propagation loss for this hybrid mode is not considered as ultralow, especially when compared to other monolithic platforms such as Si₃N₄ and lithium tantalate. Thus, we agree that our language regarding ‘ultralow’ needs to be rephrased as ‘low optical loss’. On their own, silicon nitride waveguides fabricated using the photonic Damascene process leveraged in our work do feature losses on the scale of 3-4 dB/m [Optica 5.10 (2018): 1347-1353, Nature communications 12.1 (2021): 2236]. As for low-loss LTOI PICs [Nature 629, 784–790 (2024)], they can feature losses down to 5.6 dB/m. These very low figures suggest that our reported 14.2 dB/m figure is not limited by material absorption but rather increased bending loss radiating in the unpatterned LiTaO₃ layer.

Moreover, we fully agree with the reviewer that presenting the measured insertion loss of the mode converters would offer a more complete picture of the total on-chip loss budget. Unfortunately, our current reticle design does not include dedicated test structures (e.g., cascaded tapers) that are necessary to accurately de-embed the insertion loss of a single mode converter. Without these structures, any measurement would be convoluted with other sources of loss, such as fiber-to-chip coupling. However, we believe that its insertion loss has a negligible impact on our modulation performance, as supported by our high-speed transmission experiments in the main text. To compensate for this unavailable experimental data, we provided additional simulated mode converter insertion loss data for various minimum taper width values. The results show how reducing this width with sharper lithography methods can reduce insertion losses. Such claims are also supported by our group’s prior work exploiting such methods to implement mode converters in a heterogeneous LiNbO₃-Si₃N₄ photonic integrated circuits [Nat. Commun. 14, 3499 (2023)].

Action taken: We still label losses as “ultra-low” when referring to prior work on silicon nitride waveguides fabricated with the photonic Damascene process. However, while referring to our heterogeneous platform, we now refer to these losses as “low” in the main text:

... we implement modulators that achieve low optical losses (~ 14.2 dB/m) while combining ...

... Built-in integration with thick Si₃N₄ waveguides also enables a new generation of on-chip systems leveraging both electro-optic and low loss waveguides ranging from microwave oscillators ...

In the main text (Page 3), we now also reinforce that our 14.2 dB/m likely arises from bending losses due to the smaller achieved losses in monolithic silicon nitride and lithium tantalate platforms:

This figure is larger than the 3-4 dB/m and 5.6 dB/m loss values achieved in monolithic Si₃N₄ [7, 8] and LiTaO₃ [30] circuits. This comparison provides further evidence that the hybrid waveguide’s losses do not come from intrinsic material losses but rather radiative

losses in the unpatterned and higher-index LiTaO₃ film.

Finally, we provide additional simulations pertaining to the insertion losses of the hybrid mode converter in the SI (Please see our above response to **Reviewer #1, Comment #3** in **“Supplementary information”** part).

The inclusion of a measured $V\pi L \approx 4.08$ V·cm is a strength, as it enables cross-platform comparisons without ambiguity. The text also references simulations that predict a somewhat larger $V\pi L$ for nominal geometry, which is encouraging but calls for reconciliation. A discussion aligning measured and simulated results, stating the electrooptic coefficients used, optical and microwave mode overlaps, exact electrode geometry, and layer thicknesses, would resolve -possible confusion- and reinforce confidence in the reported efficiency.

Reply: We thank the reviewer for this comment. Our original model was based on a simplified model used for bulk LNOI modulators, where the optical mode is entirely confined within the ferroelectric material. However, for our proposed hybrid LiTaO₃-on-Si₃N₄ platform, this approximation is less accurate due to the spatially varying refractive index and the optical mode across different materials. Given this limitation, we have updated our model based on perturbation theory, thereby allowing it to universally model any integrated waveguide structure (see Eqs. 2-5 in the revised SI). The simulated results using this model yields a $V\pi L$ that is in much closer agreement with our measured one, where a small deviation might be due to the fabrication tolerance. For the detailed parameters used in our model, we now list them in the Table S1 of the SI to enable readers to easily reproduce our results.

Action taken: We have revised Section 2. A in the SI. The previous simplified model for calculating the electro-optic efficiency has been replaced with the full derivation based on a perturbation theory model. We have also updated the text to discuss the agreement between the new simulation and our experimental data, and to detail the fabrication tolerances that account for the remaining deviation.

Revision to the SI (**Supplementary Section 2.A. and Supplementary Section 4.A.**):

- Eqs. (2-5): Derivation of a modified $V\pi L$ expression starting from perturbation theory.
- Table S1: List of model parameters used to calculate the reported efficiency.
- Passage in Section 4 of the SI attributing efficiency discrepancies to an uneven silicon nitride layer and a slightly varied LiTaO₃ thin film:

Based on the simulated results in Supplementary Figures 4(c,d), increasing the waveguide height within the measured range of 500 to 560 nm only slightly decreases optical loss due to stronger mode confinement in the Si₃N₄ core. This reduction comes at the cost of an increased $V\pi$ from 4.55 to 4.85 V. As for the LiTaO₃ thickness, a tens-of-nanometer variation significantly alters $V\pi$, while a thicker LiTaO₃ thin film increases metal-induced absorption loss. Therefore, we identify the uniformity of the LiTaO₃ layer as the most critical parameter for achieving consistent device performance.

Broadband S-parameter data up to ~110 GHz, including $|S_{11}|$ below -15 dB, indicate careful impedance control and low reflection at the device ports. The small-signal $|S_{21}|$ roll-off- appears consistent with ohmic attenuation ($\propto \sqrt{f}$) and modest velocity mismatch. For completeness and reproducibility, the paper should aggregate RF attenuation in dB/cm versus frequency with line length, calibration plane, and termination explicitly stated, and add S_{22} if available. Because recent LT work has highlighted the conductivity advantages of silver electrodes above 100 GHz, a short discussion of metal choice (Au vs Ag) and anticipated performance-reliability trade-offs would be valuable for readers planning to push to even higher bandwidths.

Reply: We agree with the reviewer’s suggestion that an explanation on electro-optic S₂₁ measurements is essential for completeness and reproducibility. Regarding the RF attenuation, we have updated Supplementary Figure 7(a) in the SI to show

the modulator RF loss for different device lengths (6.8 mm and 16 mm), which feature comparable values. For this measurement, the VNA was calibrated and terminated using high speed GSG probes. In the electro-optic (EO) S-parameter measurements, the CPW was terminated by a GSG probe connected to a 50 Ω coaxial load. The frequency response of a high-speed photodiode and one GSG probe were then de-embedded from the raw EO S21 signals.

Regarding the requested S22 data, we infer the reviewer is referring to the EO S22 involved in our EO measurements. In these measurements, the microwave input is port 1, and the modulated light received at the output is in port 2. Owing to the non-reciprocal features of travelling wave EO modulators, the dominant contribution to the measured EO S22 comes from co-propagating microwave reflections while sending the modulating microwave from port 2. However, this information can be deduced from the EO S21 (modulation bandwidth) and the EE S11 (impedance matching affecting microwave reflections) parameters, which universally describe the performance of travelling-wave EO modulators. Both values are already provided and analyzed in our manuscript. Adding the S22 parameter would add a level of redundancy which is not necessarily required nor typically investigated in other works.

The reviewer also raises a very important point on the metal selection for high-speed electrodes. To provide a comprehensive discussion, we have added a new analysis to Supplementary Figure 7(a), which directly benchmarks the RF loss of our Au electrodes against both Au and Ag electrodes on the monolithic LTOI platform from our recent work [Optica 11, 1614-1620 (2024)]. The updated data confirms that silver offers a slight advantage due to its lower conductivity at higher frequencies. However, our choice of gold was a strategic decision based on the trade-off between peak performance and device reliability. The chemical stability of gold electrodes ensures fabrication and device robustness, whereas silver's susceptibility to oxidation poses a significant challenge for wafer-scale manufacturing and long-term device stability.

Action taken: We have modified Supplementary Figure 7 (a) in the SI with additional RF loss data, and have updated the description in **Supplementary Section 5**.

Revision to the SI:

Supplementary Figure 7. Modulator microwave transmission and modulation efficiency. (a) Measured co-planar waveguide microwave losses on a square-root frequency axis, including the performance comparison with monolithic thin film LiTaO_3 modulators using 6 mm long gold electrodes and silver electrodes from Ref.[11]. (b) Measured transmission

of the $\text{Si}_3\text{N}_4\text{-LiTaO}_3$ modulators driven by modulation frequencies from 1 Hz to 100 kHz. Extracted modulator V_{π} value at (c) different modulation frequencies at a fixed optical 1550 nm optical wavelength and (d) different optical wavelengths at a fixed 100 Hz modulation frequency.

We first characterized the RF attenuation properties of the fabricated CPW-type modulators. Supplementary Figure 7(a) presents the extracted microwave loss of the 6.8 mm long gold electrode on the $\text{Si}_3\text{N}_4\text{-LiTaO}_3$ platform, using a 67 GHz vector network analyzer (VNA). The square-root frequency dependence of RF loss (the black dashed line shown in Supplementary Figure 7(a)) indicates the dominant ohmic loss mechanism ($\alpha \propto \sqrt{fM W}$), confirming that our fabrication techniques can effectively prevent parasitic-capacitance-induced loss [12, 13]. The measured RF attenuation is comparable to state-of-the-art ultrabroadband LTOI modulators with gold CPW electrodes [11]. Employing silver electrodes could further reduce ohmic losses, thus potentially enabling a higher EO bandwidth. However, this choice risks degrading the stability of the electrodes owing to silver's vulnerability to oxidation. Therefore, incorporating silver-based electrodes in $\text{Si}_3\text{N}_4\text{-LiTaO}_3$ modulators would require adapting them to wafer-scale fabrication and guaranteeing long-term device reliability.

The communications results are a major strength of the manuscript. Demonstrations of PAM-4 in the ~192 GBd range yielding ~333 Gb/s net after FEC, and coherent 16-QAM around 176 GBd yielding ~581 Gb/s net, establish that the devices function beyond small-signal curiosity. The setups are described clearly and the constraints of the electrical instrumentation are acknowledged (e.g., AWG bandwidth limiting the 204-GBd QPSK case). To fully enable external benchmarking, the manuscript should report per-format pre-FEC BER and EVM, the precise FEC overheads and thresholds, estimated or measured energy-per-bit based on the drive swing at the device pads, and a compact link budget that separates fiber-to-chip coupling, on-chip routing, modulator section, and any amplification used.

Reply: We thank the reviewer for the positive feedback regarding the communication results and for the constructive suggestions to add further data of our measurement results for better external benchmarking. As outlined below, we have added this information in various parts of the revised manuscript and the SI.

Actions taken: Figures 3(b) and 4(c) already provide the pre-FEC BER in the original manuscript. We now explicitly mention “pre-FEC BER” while referring to the figures and in the figure captions themselves.

Figure 3(b) provides the resulting pre-forward error correction (pre-FEC) bit error ratios (BER)...

(b) Measured pre-forward error correction bit error ratio (BER)...

Figure 4(c) shows the measured pre-FEC BER along...

(c) Measured pre-forward error correction bit error ratio (BER)

What is now Supplementary Note 8 provides the corresponding EVM for the coherent communications measurements:

Addition in the main manuscript:

In addition, the error vector magnitude (EVM) for QPSK and 16QAM signals are shown in Supplementary Figure 9(c)....

Addition in the Supplementary information:

The EVM is an alternative metric of the quality of the received signal in coherent

transmission systems and quantifies the root-mean-square (RMS) distance between the received complex modulation symbols \hat{E}_r and their ideal positions in the constellation diagram E_r [27]. We calculate the EVM normalized to the average power as Eq. (7), where N is the number of symbols over which the EVM is evaluated. Supplementary Figure 9(c) shows the resulting EVM for QPSK and 16QAM symbols after equalization as a function of the symbol rate. As expected, the EVM increases as the signal quality decreases for larger symbol rates.

Supplementary Figure 9. Extended signal quality analysis for high-speed data transmission experiments. (a) Normalized generalized mutual information (NGMI) for four-level pulse-amplitude modulation (PAM4) signals received in our intensity-modulation and direct-detection (IMDD) experiment. (b) NGMI for quadrature phase-shift keying (QPSK, brown) and 16-state quadrature-amplitude modulation (16QAM, blue) signals received in our coherent transmission experiment. All signals were transmitted using Si3N4-LiTaO3 modulators across various symbol rates. (c) Error vector magnitude (EVM) of the QPSK (brown) and 16QAM (blue) symbols extracted at the receiver for different symbol rates.

As detailed in the IMDD section, we estimate practically achievable net data rates (NDR) by comparing the normalized generalized mutual information (NGMI) estimated in our experiments to the NGMI thresholds provided in reference [46] of the main manuscript. For clarity, we added a sentence to the coherent communication section that refers to the description in IMDD section.

In addition, we added a plot of the NGMI to the supplementary (now: Supplementary Note 8) for better comparison with the thresholds given in the cited reference.

Addition in the main manuscript:

Figure 4(d) provides the corresponding AIR and NDR based on the same calculation used for the IMDD results. Additionally, Supplementary Figures 9(b,c) provides the NGMI and error vector magnitude (EVM), respectively.

Addition in the Supplementary information:

The generalized mutual information (GMI) is directly related to the achievable information rate (AIR) for bit-interleaved coded modulation and is commonly used to define performance prediction thresholds, especially for soft-decision forward error correction (SD-FEC) schemes [24]. We derive the GMI of the transmitted signals from the log-likelihood ratios (LLRs) of the received symbols based on a linear channel model with additive white Gaussian noise (AWGN) as the only impairment [25]. To compare different modulation formats, we calculate the NGMI by normalizing the GMI by the number of bits that can be encoded into a single symbol. Supplementary Figure 9(a) shows the NGMI for four-level pulse-amplitude modulation (PAM4) signals transmitted at different symbol rates in our IMDD experiment. The NGMI for QPSK and 16QAM signals transmitted in our coherent transmission experiment are depicted in Supplementary Figure 9(b). Based on these values, we select suitable forward error correction (FEC) codes with the nearest lower NGMI threshold as given in [26, Table 2] and evaluate the net data rate (NDR) by multiplying the line rates and the associated FEC code rates, see Figure 3(c) and Figure 4(d) of the main manuscript. As the symbol rate increases, the signal-to-noise-and-distortion ratio (SNDR) of the received signal decreases, and FEC implementations with larger overheads are required to enable error-free decoding. As a consequence, we

observe a reduction of the NDR beyond 192 GBd for PAM4 signals, see Figure 3(c) of the main manuscript.

Supplementary Note 8 B now provides the estimated electrical energy dissipation per payload bit based on the drive voltages at the input RF-probe. We also added a more detailed description of the electrical transmitter setup used in the IMDD experiment and added a previously missing RF amplifier in the setup in Figure 3(a):

Addition in the main manuscript:

... We drive the MZM with electrical PAM4 signals, that are synthesized by offline digital signal processing (Tx-DSP) based on pseudo-random bit sequences (PRBS) and root-raised cosine (RRC) pulse-shaping filters [30], and that are converted to the analogue domain using a high-speed arbitrary waveform generator (AWG, M8199B, Keysight Technologies Inc.). A 20 cm-long RF cable followed by a broadband RF amplifier (SHF T850 B, SHF Communication Technologies AG) sends the drive signals to the coplanar transmission line of the MZM via a first impedance-matched probe in a ground-signal-ground configuration. A second probe terminates the transmission line with a 50 Ω coaxial termination. ...

Figure 3. Transmission experiments using intensity-modulation and direct detection (IMDD) (a) Experimental setup. ECL: External cavity laser; FPC: Fiber polarization controller; AWG: Arbitrary-waveform generator; Tx-DSP: Transmitter-digital signal processing (offline); Amp.: RF amplifier; EDFA: Erbium-doped fiber amplifier; BPF: Bandpass filter; VOA: Variable optical attenuator; PD: Photodiode; RTO: High-speed real-time oscilloscope (256 GSa/s, 105 GHz); Rx-DSP: Receiver-digital signal processing (offline). (b) Measured pre-forward error correction bit error ratio (BER) versus symbol rate for PAM4 transmission, with horizontal dashed lines indicating the thresholds for soft-decision forward error correction with 15 % and 25 % coding overhead (15 % SD-FEC, 25 % SD-FEC) and for hard-decision FEC with 7 % coding overhead (7 % HD-FEC). The eye diagrams of two selected data points A and B are shown in panels (d) and (e). (c) Achievable information rates (AIR, orange dashed line) and corresponding net data rates (NDR, solid brown line) of the IMDD measurements, showing that the highest achieved NDR is 333 Gbit/s using a PAM4 signals at a symbol rate of 192 GBd. (d), (e) Eye diagrams (left) at selected symbol rate of 160 GBd PAM4 (d) and 200 GBd PAM4 (e), labeled by A and B in panel (b), along with the associated histograms (right) of the reconstructed signal amplitudes in the center of the corresponding symbol slot.

Addition in the Supplementary information:

To estimate the electrical energy dissipation per transmitted bit, we first need to extract the voltage levels that were fed to the modulator during the transmission experiment. To this end, the electrical transmitter was directly connected to the real-time oscilloscope (RTO,

UXR 1004A, Keysight Technologies Inc.) and operated with the same parameters and signals as in the optical transmission experiment.

In case of the IMDD experiment, the transmitter consisted of a high-speed arbitrary-waveform generator (AWG, M8199B, Keysight Technologies Inc.) and a subsequent RF amplifier (SHF T850 B, SHF Communication Technologies AG), which was directly connected the RF probe during the transmission experiment. The AWG and the RF amplifier are connected by a 20 cm-long RF cable. For measuring the voltage levels that were fed to the modulator during the transmission experiment, we connect the output of the RF amplifier directly to the RTO. To estimate the electrical energy dissipation during the data transmission experiment, we calculated the average electrical power fed to the modulator from the voltage samples recorded by the RTO. Assuming impedance matching between the terminated modulator and the internal resistance of the transmitter ($R = 50 \Omega$), the average electrical power is calculated from the instantaneous voltage $u(t)$ by Eq. (8), where T is an observation time that covers many symbols ($T = 32.8 \mu\text{s}$ in our experiments). The electrical energy dissipation per transmitted payload bit is given by dividing the average electric power by the NDR Eq. (9). Note that the transmission experiments at the various symbol rates were conducted with a constant drive-voltage swing applied to the MZM, which was ensured by re-adjusting the output voltage settings of the AWG for each symbol rate as to maintain the voltage difference between the highest and the lowest PAM4 signal level in the range $(1.25 \pm 0.05) \text{ V}$. Supplementary Figure 10(a) shows the net energy per bit for transmitted PAM4 signals at various symbol rates. Since the drive voltages were kept constant, the energy dissipation follows the trend of the inverse NDR, see Figure 3(c) of the main manuscript. The lowest energy dissipation per payload bit amounts to 12.9 fJ bit^{-1} , reached at a symbol rate of 176 GBd . Non-ideal voltage settings lead to smaller deviations from of the measured values from a continuous trend, as indicated by a dashed line in Supplementary Figure 10(a).

For the coherent data transmission experiments, the drive voltages applied to both MZMs must be considered individually, since the MZM in our sample device turned out to have different half-wave voltages. Due to the limited physical space between the two input connectors at the feeding RF-probe, the electrical transmitter configuration differs from that one used in the IMDD setup. Specifically, two outputs of the AWG were each directly connected to separate broadband RF amplifiers, followed by two broadband bias-tees and two 20 cm-long RF cables. The components in both signal paths are nominally identical. In this experiment, we kept the voltage settings of the AWG constant, leading to a decrease of the voltage swings coupled to the MZM with increasing symbol rates. As in the IMDD case, the voltage levels for the transmitted QPSK and 16QAM symbols are extracted by connecting the RF cables to the input of the RTO, and the average electrical power and the electrical energy dissipation per transmitted payload bit were estimated in analogy to Equations 8 and 9. For the MZM transmitting the in-phase signal, the drive voltage difference between the lowest and the highest signal level decreases from 0.8 V at 144 GBd to 0.4 V at 204 GBd . For the quadrature MZM, the voltage swing reduced from 1.6 V to 0.8 V over the same symbol rate range. The electrical energy per transmitted payload bit is depicted in Supplementary Figure 10(b). The minimum net electrical energy dissipation per bit amounts to 9.4 fJ bit^{-1} and was achieved for 16QAM signals with a

symbol rate of 176 GBd.

Supplementary Figure 10. Electrical energy efficiency analysis for high-speed data transmission experiments. (a) Estimated electrical energy dissipation per payload bit calculated from the drive voltages measured for four-level pulse-amplitude modulation (PAM4) signals using intensity-modulation and direct-detection (IMDD) and (b) for quadrature phase-shift keying (QPSK, brown) and 16-state quadrature-amplitude modulation (16QAM, blue) signals. A third-order polynomial fit (dashed line) provides a guide to the eye. All quantities are plotted for different symbol rates. Please note the different scales of the vertical axes.

Now, we also provide characterization metrics that give a more quantitative picture of our compact link budget. Optical losses due to fiber-to-chip coupling, on-chip routing, and the modulator section are now provided in **Supplementary Notes 3,4**, as addressed in **Reviewer #1, Comment #3 in the “Supplementary information” part** and **two comments down in this reply**. Finally, as mentioned in the main text on **page 6**, we do use optical amplification in our experiments.

After modulation, an EDFA amplifies the optical signal from -13 dBm to 16 dBm and a bandpass filter subsequently reduces out-of-band spontaneous emission noise. At the receiver, the modulated signal enters a 90° optical hybrid module, where it interferes with a continuous-wave local oscillator provided by a second ECL to down-convert the optical signal to four electrical baseband signals.

The introduction fairly motivates LT over LN with lower birefringence and improved photostability and DC--bias stability. However, these advantages are not yet quantified on the presented platform. Including time-resolved bias drift under realistic optical powers and over a relevant temperature range would turn a qualitative motivation into a platform-level evidence point. Even a modest A/B comparison versus a representative -LN--on-- Si_3N_4 control device, if available, would strengthen the case materially.

Reply: We thank the reviewer for his/her constructive suggestion on quantifying the DC-bias stability of our integrated hybrid MZMs. In the revised SI, we present a dedicated experiment characterizing the device's long-term stability. This was realized by an in-house photonic package to isolate the intrinsic bias drift of our demonstrated platform from external factors such as chip-to-fiber coupling efficiency drift due to unstable mounting on flexure stages. A low optical power shift of less than 0.5 dB over one hour showcases its superior performance compared to state-of-the-art monolithic LiTaO_3 devices [Optica 11, 1614-1620 (2024), Optics Express 32, 44115 (2024)].

Regarding the additional suggestion, a temperature-dependent measurement is valuable but presents a significant challenge to our current packaging technique. Specifically, our current epoxies gluing the fibers to the chip can exhibit thermal stress that can damage the device while heating it to temperature ranges relevant for such measurement. Similarly, a comparison with

a LN-on-Si₃N₄ device was not possible as we do not have any available devices at this time. Nevertheless, we believe that the added long-term stability data provides quantitative evidence for one of the key advantages of our heterogeneous platform and directly addresses the core concern raised by the reviewer.

Action taken: We have updated the main text (**Figure 2**) and the SI (**Supplementary Section 6**), which includes the result and discussion of the DC bias stability measurement. To avoid repetition, the added section details can be inferred from our above response to **Reviewer #1, Comment #3 in the “main text” part**.

The text describes fiber--to-chip coupling on the order of several dB per facet and indicates the need for amplification to meet receiver power targets in some experiments, which is entirely plausible for -early-stage devices. Nevertheless, a clear separation of -external-coupling loss from intrinsic -on-chip insertion loss of the MZM and IQM would meaningfully assist practitioners. Reporting measured -mode-converter loss, -on-chip device IL, and facet coupling with uncertainties would also help reconcile the -communications-experiment power budgets with the -passive-loss- measurements.

Reply: We thank the reviewer for this constructive suggestion. We have performed the insertion loss (IL) measurements for modulators for five fields in our manufactured wafer. To extract the MZM device IL, our method was to first characterize the reference LiTaO₃-Si₃N₄ hybrid waveguide (fiber - edge taper - transition taper - waveguide - transition taper - edge taper - fiber) to determine the fiber coupling of this chip. We then measured the total loss of the MZM device and subtracted the reference coupling loss to extract the intrinsic on-chip MZM IL.

From the measured results shown in the SI, our current device reveals an unexpectedly high IL (in excess of several tens of dB), however it did not affect the modulation efficiency a lot. We have identified the cause as a non-optimal DUV dose for defining the Si₃N₄ pattern. This results in a partially exposed taper shape, affecting both the edge coupler and on-chips components like Y-splitters. To explain that, we have also included the results measured from a previous sample with an identical optical design, which exhibits excellent performance with a total IL of a few dB. However, these old samples suffer from a prominent plasma-induced loss introduced by the resist strip machine (Tepla 300).

Therefore, the high loss observed on the current wafer is a process-specific issue that can be rectified in future fabrication runs. With a properly optimized fabrication process, we are confident that our platform is highly promising for achieving high speed communication with low power budgets.

Action taken: We have added a new table (**Supplementary Table 2**) to the SI (**Supplementary Section 3.B.**) presenting the detailed loss budget analysis:

Supplementary Table 2. Insertion loss (IL) extraction for 6.8 mm MZMs

Field	Taper-to-taper IL (dB)	Device number	Total IL (dB)	MZM IL (dB)
F1 (packaged)	/	Device 103	-23.0000	/ (packaged)
F3	-20.0736	Device 101	-35.5309	-15.4573
F3	-20.0736	Device 102	-29.6210	-9.5474
F3	-20.0736	Device 103	-25.904	-5.8304
F5	-26.3650	Device 101	-44.8847	-18.5197
F5	-26.3650	Device 102	-37.2224	-10.8574
F5	-26.3650	Device 103	-39.7224	-13.3574
F9	-25.7133	Device 101	-45.7785	-20.0652
F9	-25.7133	Device 102	-46.4975	-20.7842
F9	-25.7133	Device 103	-44.1042	-18.3909
Old sample	-6.8627	Device 101	-11.986	-5.1233
Old sample	-6.8627	Device 102	-11.5656	-4.7029

Old sample	-6.8627	Device 103	-14.6682	-7.8055
------------	---------	------------	----------	---------

We have also added a new paragraph to discuss these results, explaining the fabrication anomaly on this fabrication run and highlighting the intrinsic low-loss potential proven by previous samples:

Regarding the insertion loss, Supplementary Table 2 details the insertion loss (IL) budgets for the 6.8 mm long single MZM devices. The “Taper-to-taper IL” represents the total fiber-to-chip coupling loss measured from a reference straight waveguide with two mode transition tapers, while the intrinsic “MZM IL” is obtained by subtracting the coupling loss from the “Total IL” of the modulator device. Missing data points in the table, such as F5, are due to some devices being damaged during handling or shipped away for high speed telecommunication testing. The MZM devices from the primary wafer exhibit a high IL near 20 dB as well as a total IL near 40 dB. This is attributed to a non-ideal DUV lithography exposure dose during the Si₃N₄ fabrication step, which resulted in malformed taper structures for both the edge couplers and the on-chip Y-splitters. To demonstrate the platform’s intrinsic low-loss capability, we include data attributed to chips labeled as “Old sample” from our group’s prior work. The devices feature identical optical designs with successfully patterned tapers. However, their high-frequency performance is uniformly degraded by parasitic capacitance loss, which we attribute to the high-power oxygen plasma used during the photoresist removal step. This sample shows a competitive coupling loss of near 3.4 dB per facet and a reasonable on-chip MZM IL of 5-8 dB, which confirms that the high loss on the current fabrication run is a correctable anomaly and not limited by optical design.

Overall methods transparency is good and likely, nevertheless the paper would still benefit from explicit uncertainty estimates for $V_{\pi L}$, loss extractions, and bandwidth fitting; precise VNA calibration details and de-embedding steps; and, where applicable, references to public PD-K-like parameters (layer thicknesses, refractive indices at test wavelengths). Adding micrographs or cross-sections of critical transitions (e.g., vertical adiabatic converters-) would also round out the reproducibility story.

Reply: We agree that including further details on our modeling and data processing would benefit future studies attempting to reproduce our results. As specified below, we have added the information requested by the reviewer. We have uploaded the code de-embedding the response of our apparatus from that of the modulator in a data sharing repository (<https://doi.org/10.5281/zenodo.17276861>) which is now linked to our manuscript. However, regarding the performance uncertainty, we instead present wafer-level measurements for modulation efficiency and device loss, as mentioned in the above replies, instead of reporting the often negligibly small standard errors from conventional curve fits (e.g., for $V_{\pi L}$ or resonator intrinsic linewidth). We believe this approach more efficiently conveys uncertainties related to the performance of our platform.

Action taken:

We have added the description of VNA calibration and de-embedding procedure to the main text (**Page 5**):

A standard calibration kit first ensures two-port calibration by setting the reference plane at the tips of the two high-speed electrical probes. The EO measurement then relies on a modification in the setup where the VNA’s output connects to the high-speed PD detecting optical output rather than the GSG probe. De-embedding independently characterized frequency responses from the PD and one GSG probe finally yields the intrinsic EO response of the modulator.

PDK-like parameters (layer thicknesses, refractive indices at test wavelengths) are now

provided in Supplementary Table 1 of the SI (**Supplementary Section 2.A**):
Supplementary Table 1. Parameter values used to model electro-optic performance.

Parameter	Description	Value
λ	Optical Wavelength	1.55 μm
n_{eff}	Effective refractive index	1.828
ϵ_{Air}	Relative permittivity of air (defines $\epsilon_{11}, \epsilon_{22}, \epsilon_{33}$)	$(1)^2$
ϵ_{SiO_2}	Relative permittivity of SiO_2 (defines $\epsilon_{11}, \epsilon_{22}, \epsilon_{33}$)	$(1.444)^2$
ϵ_{SiN}	Relative permittivity of Si_3N_4 (defines $\epsilon_{11}, \epsilon_{22}, \epsilon_{33}$)	$(1.992)^2$
ϵ_o	Ordinary relative permittivity of LiTaO_3 (defines $\epsilon_{11}, \epsilon_{22}$)	$(2.123)^2$
ϵ_e	Extraordinary relative permittivity of LiTaO_3 (defines ϵ_{33})	$(2.119)^2$
r_{13}	13 component of LiTaO_3 's Pockels tensor	8.4 pm/V
r_{33}	33 component of LiTaO_3 's Pockels tensor	30.5 pm/V
$E_{\text{DC}}(r)/V$	DC field to applied voltage ratio	6 μm
	Si_3N_4 waveguide dimensions	1 $\mu\text{m} \times 0.5 \mu\text{m}$
	LiTaO_3 film thickness	300 nm
	Interlayer oxide thickness	30 nm

In **Supplementary Section 4** of the SI, Supplementary figure 6(d) now provides a top-view SEM of the vertical adiabatic converter. We abstained from taking a cross-section of the device to preserve it for future studies:

As shown in Supplementary Figure 6(c), the 500 nm minimal width of the taper primarily limits transmission. The patterning of the LiTaO_3 layer relied on a mask-less lithography method. As shown in the scanning electron micrograph of Supplementary Figure 6(d), this type of lithography produces coarser features resulting in greater insertion losses. The simulated data provided in Supplementary Figure 6(c) suggest that decreasing this width down to 200 nm can decrease the taper's insertion loss down to 0.08 dB. As previously demonstrated for heterogeneously integrated $\text{LiNbO}_3\text{-Si}_3\text{N}_4$ circuits, DUV optical lithography can reach these widths and insertion loss figures.

Supplementary Figure 6 Simulated transmission of silicon nitride-lithium tantalate waveguide transitions. (a) Schematic diagram and corresponding FDTD simulations of the adiabatic coupling from the Si_3N_4 waveguide to the hybrid $\text{Si}_3\text{N}_4\text{-LiTaO}_3$ waveguide. (b) Simulated transmission spectrum of the waveguide transition for various minimum LiTaO_3

taper widths. (c) Simulated electric field distribution of the adiabatic taper for a 500 nm minimum LiTaO₃ taper width. (d) Top-view scanning electron micrograph of the waveguide transition where the patterned LiTaO₃ layer features blue false colors.

Monolithic LTOI platforms have reported low propagation loss and strong efficiency at ~40-GHz-class bandwidths; discrete thin-film LT modulators have reached beyond 100 GHz; heterogeneous LN-on-Si₃N₄ has already proven the architectural value of marrying a strong Pockels material to ultra-low-loss routing. This manuscript's unique contribution is to combine the LT material with Si₃N₄ at wafer scale and then validate with both IM/DD and coherent communications at symbol rates competitive with the best contemporary demonstrations. Given the sustained interest in coherent transceivers and RF-photonics, the practical impact is credible, particularly if the authors add the benchmarking details outlined above and as such the manuscript could be accepted for publications after the requested revisions have been added.

-

Reply: We appreciate the positive feedback and insightful suggestions from the reviewer. We have addressed all the points raised and sincerely hope that the point-by-point response meets the reviewer's expectations for publication. As the reviewer noted, this work validates a photonic architecture by combining the advantages of ferroelectrics and CMOS-compatible silicon nitride. We believe this is a pivotal step towards next-generation photonic systems.

Reviewer #3 (Remarks to the Author):

This is a very good paper which addresses one of the current largest bottlenecks associated with high speed data transmission. Silicon and lithium niobate based optical modulators are areas of active research and the competing technologies; Single lane speeds are limited by modulator speeds and therefore, companies and researchers are intensely studying ways to achieve better on-chip modulators. In this paper authored by Cai et. al, they introduce lithium tantalate based modulators which are integrated with ultra-low loss silicon nitride waveguide devices. I recommend publishing this paper and have some suggestions for the authors:

Reply: We sincerely thank the reviewer for the acknowledgement of our work and for the efforts to further improve our manuscript. As the reviewer noted, the hybrid photonic platform is of great interest to the photonic community. We believe that our work can set a new benchmark for the performance of heterogeneous integrated systems for practical applications, such as high-speed telecommunication.

1) The reported V_{π} is 6V. Can the authors comment on how this compares with other state of the art modulators? A commonly used figure of merit is $V_{\pi}L$. A table which compares this figure of merit, bandwidth and other commonly used performance metrics with other modulators would be great for readers.

Reply: We thank the reviewer for this excellent suggestion. Comparing our results with other state-of-the-art modulators can provide a more intuitive understanding of the performance of our demonstrated heterogeneous platform, and a comprehensive benchmarking table is one of the most effective ways to present above information clearly.

Action taken: Following the reviewer's suggestion, we have added a new section in the SI (**Supplementary Section 9**) that provides a detailed comparison of our work with other leading modulator technologies. A comparison table is also attached to compare their key figures of merit, which includes the modulation bandwidth, the modulation efficiency, data rate and BER.

To avoid repetition, the added table and discussion can be accessed from our above response to **Reviewer #2, Comment #1**.

2) The authors demonstrate the modulator platform with both IMDD and coherent data transmission. Can they comment on whether the highest rates demonstrated were limited by the equipment or the modulator platform itself?

Reply: Regarding the bottleneck of the achievable rate using our $\text{LiTaO}_3\text{-Si}_3\text{N}_4$ modulators, we would like to offer the following discussion from two perspectives:

1. Device limitation: The data rate of the modulator is limited by modulation bandwidth. From the measured results, our 6.8 mm modulator exhibits a remarkably flat EO S_{21} response up to 110 GHz and low microwave reflections in the electrical-to-electrical S_{11} signals, indicating excellent velocity matching and impedance matching. Thus, we are confident that the modulator platform itself is not the primary obstacle for us to achieve a higher data transmission rate. The slight roll-off is due to the intrinsic ohmic loss of the gold electrodes. While this performance reaches state-of-the-art values (please see the comparison provided in the updated Supplementary figure 7(a) in the SI), ohmic losses could be improved by employing a more conductive metal, such as silver, thereby pushing a higher data rate.
2. Measurement limitation: Currently, the highest data rates in our transmission experiments were significantly constrained by the bandwidth of our electrical link. As illustrated in our main text: "Here, the 75 GHz electrical 3 dB-bandwidth of the AWG [46] mainly limits the performance of this experiment.", the primary bottleneck is our arbitrary waveform generator (AWG, Keysight M8199B), which greatly attenuates high-frequency signals and degrades signal quality before it entering the CPW-type electrode. Meanwhile, the entire

electrical path, including all cables and probes, contributes to the total bandwidth limit of the telecommunication system.

This "Electronic Bottleneck" is a well-known challenge in the industry, but developing ultra-high-bandwidth AWGs exceeding 100 GHz is extremely costly. Our demonstrated IQ modulator (Fig.4 in the main text) overcomes this bandwidth limited data transmission rate, thereby allowing us to double the data rate without requiring higher electrical bandwidth. Moreover, the industry's roadmap towards 1.6 Tb/s could be realized by parallel telecommunication channels with wavelength-division multiplexing, particularly by combining multiple modulators with an on-chip array waveguide grating to construct monolithic optical transceivers. However, this requires more efforts on dedicated designs and advanced fabrication technique, which represents a key direction for our future work.

These discussions mostly apply to the MZM used in the IMDD experiment. The IQ modulator used for coherent communications had a bandwidth of 34 GHz. Therefore, improved group velocity matching consists of the best path to increase our coherent data rates.

Action taken: Our results section already mentions that the equipment (specifically the AWG) primarily constrains our IMDD rates using our highest bandwidth modulators. However, we now re-emphasize this point at the end of the Data Transmission part of the Results section.

Revision to the main text (Page 8):

Achieving greater data rates will likely require specialized driver electronics given the Si_3N_4 - LiTaO_3 modulator's >100 GHz bandwidth.

For reference, we have also added the bandwidth of the IQ modulator in Supplementary Table 3.

3) Because of the lack of an easily available direct bandgap material in CMOS, III-V continues to be relevant. In the discussion section, the authors discuss III-V active materials and their potential integration with the lithium tantalate- Si_3N_4 platform in the future through bonding. Can the authors comment on the difficulties that may be associated with the adoption of this in CMOS foundries.

Reply: We thank the reviewer for this question. To clarify, the core of our strategy is the use of wafer bonding to overcome the incompatibilities between different materials and standard CMOS foundries. The primary difficulty in the adoption of III-V active materials is the contamination they add to processing. Thus, the monolithic growth or processing of III-V thin film is strictly prohibited within a CMOS foundry. Regarding its incompatibility with the demonstrated process flow for LiTaO_3 -on- Si_3N_4 platform, the high temperature required of epitaxy III-V growth, such as MOCVD and MBE, usually far exceeds the Curie point of lithium tantalate (~610–700 °C). Based on the same strategy in our work, our planned process flow for full co-integration is: (1) CMOS-foundry fabrication (e.g., Photonic Damascene process or subtractive process) for the silicon nitride passive photonic circuits. (2) Wafer-to-wafer bonding: the optical-grade LTOI wafer and III-V epitaxial wafers are prepared on separate substrates, and the same wafer-scale bonding demonstrated in our work can be utilized to sequentially transfer the LiTaO_3 thin film and III-V thin film onto the Si_3N_4 wafer. (3) Process for patterning and metallization for the LiTaO_3 and III-V devices in a separate processing line. This post-process effectively isolates the CMOS-incompatible materials and processes, avoiding any risk of cross-contamination. Although die-to-wafer bonding has been demonstrated for similar integration tasks [Nature 620, 78–85 (2023).], our proposed approach would feature excellent wafer homogeneity and high-volume manufacturing, which is more scalable for co-integrating III-V active devices within foundry-processed silicon photonics.

Action taken: We have modified the description of co-integration of III-V thin film onto our platform in the Discussion section to highlight the reason for the lack of III-V materials in CMOS foundries.

Revision to the main text (**Page 8**):

Our $\text{Si}_3\text{N}_4\text{-LiTaO}_3$ platform could potentially benefit from additional functionalities brought by platform extensions. Namely, adapting its bonding process could lead to the integration of III-V active optical materials [55–58] and potentially to a multilayer platform featuring both III-V components [59] and $\text{Si}_3\text{N}_4\text{-LiTaO}_3$ hybrid waveguides. This wafer-scale integration approach offers a scalable solution that avoids the risk of cross-contamination inherent to processing III-V materials within a complementary CMOS foundry.

4) In a similar vein, lithium tantalate is not currently widely used in CMOS foundries. Some discussion about what could accelerate adoption in foundries would be useful.

Reply: We would like to thank the reviewer for his/her suggestion. The challenge facing the adoption of lithium tantalate in CMOS foundries is quite similar to that of III-V materials, which can introduce metallic ion contamination (such as lithium ions) into a standard CMOS fabrication line. Therefore, our response to this question is the same as the core strategy in our work: The most accessible approach is not to force the material into the standard foundry, but rather to integrate it onto the CMOS-processed platform via wafer bonding in a separate process line. We believe this method would provide a scalable pathway for combining the advantages of both platforms.

Action taken: We have modified the description in the introduction part to clarify the current challenge and frame our work as a solution, which is its co-integration with CMOS-compatible platforms.

Revision to the main text (**Page 1**):

Improved economies of scale for lithium tantalate substrates due to their role in 5G/6G RF filters [33] further encourages the use of this material in integrated modulators. At the current stage, heterogeneous integration of this material offers an immediately accessible route towards leveraging its features in high-volume manufacturing platforms such as silicon nitride.

Reviewer #4 (Remarks to the Author):

The authors achieved a wafer-scale heterogeneous bonding of LiTaO₃ onto Si₃N₄ PIC platform. Compared to monolithic electro-optic photonics platforms, this approach leverages the low-loss Si₃N₄ waveguides while simultaneously exploiting the electro-optic properties of LiTaO₃. The work demonstrates EO modulation with a $V\pi$ of 6 V and bandwidths up to 100 GHz, as well as both IMDD and coherent signal transmission, showing performance comparable to EO modulators realized on other PIC platforms. The manuscript is suitable for publication, but the authors should address the following points before acceptance:

We thank the reviewer for the appreciation of our work and our results. We have addressed his/her constructive comments carefully in our response below.

1. Compared to state-of-the-art EO modulators, this work exhibits relatively low modulation efficiency, with a reported $V\pi\cdot L$ of 4.08 V·cm. For the MZM used in coherent data transmission, the electrode length is up to 13.5 mm. What is the underlying reason for this low efficiency? Is it due to limited optical mode overlap or insufficient electric-field overlap with the active LiTaO₃ film?

Reply: We would like to clarify that the origin of the relatively low $V\pi\cdot L$ arises from the heterogeneous nature of our platform. Given the simulation results in the SI, we do believe that the modulation efficiency is primarily limited by the optical mode distribution within the lithium tantalate film (approximately 50%), rather than by the strength of the microwave-optical interaction owing to the design of the electrode gap. A lower $V\pi\cdot L$ can be easily realized by reaching approximately 100% mode overlap in monolithic ferroelectric waveguides. Future generations of our heterogenous platform could incorporate more advanced designs, such as three-dimension distinct waveguide structures coupled with adiabatic tapers, as discussed in our Discussion section.

Action taken: The first paragraph of our Discussion section already mentions tradeoffs between propagation losses and modulation efficiency. However, we have added additional simulations in the SI (**Supplementary Section 3.A.**) illustrating how the thickness of the hybrid waveguide's silicon nitride layer and lithium tantalate thin film affects both propagation losses and our $V\pi\cdot L$ value. These can be found in Supplementary Figure 4 as mentioned in our previous response to **Reviewer #1, Comment #2 in the "main text" part.**

2. Was the EO MZM measured under a DC bias? If so, was any bias drift observed, and how severe or minor was it compared to that in lithium niobate devices?

Reply: The question of the device stability under a DC bias is indeed an important one that should be discussed in our work. To quantitatively characterize the DC bias drift, we first set the MZM to its quadrature point using a DC voltage. We then monitor the optical output power in a free-running state for one hour. To ensure the intrinsic device stability, the modulator was packaged so as to minimize environmental fluctuations attributed to fiber-to-chip coupling. The measured power drift, observed to be less than 1 dB, is comparable to previous results based on a monolithic lithium tantalate platform [Opt. Express 32, 44115 (2024)], and is superior to the results in lithium niobate devices [Nat. Commun. 11, 3911 (2020)].

Action taken: In the main text, we have added a new paragraph and updated **Figure 2** to show our DC bias measurement. We also added a new section to the SI (**Supplementary Section 6.**), including the image of the packaged LiTaO₃-Si₃N₄ modulator and details about this packaging method.

To avoid repetition, the added section details can be inferred from our above response to **Reviewer #1, Comment #3 in the "main text" part.**

3. In line 251, some works in other heterogeneous platforms are cited (mainly LiNbO₃ or LiTaO₃). However, there have also been similar demonstrations using other EO materials, such as barium titanate or SiN PIC platforms, which provide comparable integration advantages but with significantly larger Pockels coefficients. These works should also be discussed in comparison. In addition, including other types of EO modulators and summarizing performance metrics (both monolithic and heterogeneous) in a comparison table would make the advantages of this heterogeneous Si₃N₄-LiTaO₃ platform over other photonic platforms clearer.

Reply: We thank the reviewer for bringing up this point. As addressed in comments raised by other reviewers, we have added a comparison table in what is now section 9 of the SI which includes the PIC platforms mentioned by the reviewer.

Action taken: We added **Supplementary Section 9** and **Supplementary Table 3** to our work. To avoid repetition, the table details can be seen from our above response to **Reviewer #2, Comment #1**.

4. The advantages of LiTaO₃ over lithium niobate in this context of modulators were not really very clear to me – lower birefringence and reduced photorefractive effects were mentioned, but how significant is the advantage (especially if we were to compare LN-with-nitride and LT-with-nitride side by side)? I suppose the “fast tunable LIDAR” application makes sense, but a more quantitative discussion would strengthen the motivation.

Reply: We would like to thank the reviewer for prompting this discussion on the motivation for using thin film lithium tantalate in this work. Compared to lithium niobate, lithium tantalate does not necessarily achieve a lower $V\pi\cdot L$ or a higher bandwidth. Both materials feature similar metrics. However, lithium tantalate achieves long-term modulation stability, large-volume manufacturing and less device anisotropy:

1. Less photorefractive effect:

The photorefractive effect is primarily driven by iron impurities in the crystal. Congruent LiTaO₃ is known to be more resistant to this photorefractive damage than congruent LiNbO₃ due to fewer defects introduced during its production process [Nature 629, 784–790 (2024)], thereby enhancing its power handling capability for modulators [ACS Photonics 10.1021/acsp Photonics.5c00159 (2025)]. Meanwhile, this intrinsic material advantage contributes directly to a superior DC bias stability, as confirmed by our long-term stability measurement in the SI. For applications requiring a stable operation or high optical power (like analog links or LIDAR), this is a decisive advantage, as it is promising to reduce or even eliminate the need for complex and power-hungry electrical feedback loops.

2. Volume manufacturing:

Driven by its widespread applications in RF filters for 5G/6G applications, the ecosystem for LTOI wafers is much more mature and has been at a stage of large-volume manufacturing. Thus, this established supply chain ensures that our LiTaO₃-Si₃N₄ PICs can benefit from cost-effective and reliable production compared to lithium niobate.

Regarding the Reviewer's suggestion on fast tunable LIDAR, we fully agree that this is a compelling application that benefits from our platform's excellent modulation efficiency and DC operation stability. However, a full LIDAR demonstration is a system-level undertaking, which is beyond the scope of this current device-focused manuscript. We believe our new long-term DC bias drift measurement (now included in Figure 2 and the SI, **please see the above Comment #2**.) quantitatively substantiates the advantage of using lithium tantalate. The superior stability we obtained shows intrinsically lower photorefractive effects in the unetched lithium tantalate thin film.

Reviewer #5 (Remarks to the Author):

Reply: We thank the reviewer for their involvement in Nature Communications co-review process and hope that the comments they raised in the main reviewer's referral were suitably addressed.

A point-by-point response to the reviews' comments of the manuscript "Heterogeneously integrated lithium tantalate-on-silicon nitride modulators for high-speed communications"

We would like to express our sincere gratitude to all the reviewers for their detailed review and insightful suggestions regarding our revised manuscript. Here, we provide point-by-point answers to the reviewers' comments.

Reviewer #1 (Remarks to the Author):

The authors replied my comments and revised the manuscript properly. The manuscript can be published in Nature Communication. In supplementary figure 5.c, the unit of x-axis is incorrect. it should be MHz.

Reply: We are glad to hear that our revisions addressed the reviewer's comments and thank the reviewer for pointing out the typographical error from Supplementary Figure 5.c. In our revised manuscript, we replaced the incorrect 'THz' label by the correct 'MHz' label.

Reviewer #2 (Remarks to the Author):

In the updated manuscript and supplementary information, the authors have addressed my concerns. One last comment would be to include the device insertion losses in the new Supplementary Table 3.(Performance metrics of high-speed EO modulators). In my opinion, the innovation and engineering values of the work are now good for the publication.

Reply: We are glad to hear that our revisions addressed the reviewer's comments. As suggested by the reviewer, Supplementary Table 3 now lists device insertion losses.

Reviewer #3 (Remarks to the Author):

The authors have addressed my comments. I recommend publishing the manuscript.

Reply: We are glad to hear that our revisions addressed the reviewer's comments.

Reviewer #4 (Remarks to the Author):

I'm satisfied by the replies to the points I raised in my earlier review.

Reply: We are glad to hear that our revisions addressed the reviewer's comments.